

# Quantitative reconstruction of precipitation changes in the Iberian Peninsula during the Late Pleistocene and the Holocene

Liisa Ilvonen[1,2], José Antonio López-Sáez[3], Lasse Holmström[4], Francisca Alba-Sánchez[5], Sebastián Pérez-Díaz[3], José S. Carrión[6], Heikki Seppä[1]

[1]Department of Geosciences and Geography, University of Helsinki, P.O. Box 64, FI-00014 University of Helsinki, Finland
[2]Department of Mathematics and Statistics, University of Helsinki, P.O. Box 64, FI-00014 University of Helsinki, Finland
[3]Instituto de Historia, CSIC, c/ Albasanz, 26-28. 28037 Madrid, Spain
[4]Research Unit of Mathematical Sciences University of Oulu, P.O.Box 8000, FI-90014 University of Oulu, Finland
[5]Department of Botany, University of Granada, 18071 Granada, Spain
[6]Department of Plant Biology, Faculty of Biology, University of Murcia, 30100 Espinardo, Murcia, Spain

*Correspondence to*: Liisa Ilvonen (liisa.ilvonen@helsinki.fi)

**Abstract.** Precipitation is a key climate driver of vegetation and ecosystems of the Iberian Peninsula. Here, we use a regional pollen-climate calibration model and fossil pollen data from seven sites from different parts of Spain to provide quantitative reconstructions of annual precipitation values for the last 15,000 years. Our records show that in the Late Pleistocene (~15,000 to 11,600 cal yr BP) precipitation changes took place markedly in tune with the temperature trends in northern Europe, with higher precipitation during the Greenland interstadial 1 (Bølling-Allerød) and lower precipitation during the Greenland stadial 1 (Younger Dryas). The early Holocene was characterized by a rapid precipitation increase after 11,600 cal yr BP, followed by a slowly declining trend until roughly 8000 cal yr BP. From 8000 to 4000 cal yr BP the reconstructed precipitation values are the highest in most records, with maximum values nearly 100 % higher that the modern reconstructed values. The results suggest a gradually declining precipitation over the last four millennia, although the late-Holocene reconstructions are biased by intensifying human impact on vegetation. In general, our results suggest that the main changes in precipitation in the Iberian Peninsula have occurred in pace with the main temperature changes in the North European-Atlantic region, with warm (cold) periods in the North corresponding with humid (dry) periods in the Iberian Peninsula.

## 1 Introduction

Successful use of quantitative transfer functions for climate reconstructions from pollen and other biological proxy data have many requirements. Of particular importance is that the reconstructions must be focused on regions where the palaeorecords are climatologically sensitive to the climate variable of interest and where it is possible to construct high-quality modern calibration sets (Birks, 1995). Within the scope of pollen-based climate reconstructions, such regions are where there exists a simple zonal climatic gradient, determined, or strongly influenced, by one or few dominant climatic variables, and where there exists equally clear vegetation zonation determined by these dominant climatic variables (Seppä et al., 2004). The



Mediterranean region is one of such regions, as it is a climatic transition area between the Atlantic to Mediterranean and subtropical to middle latitude climate gradients, and displays wide regional climate variability and large gradients, especially following a North-South transect (Karagiannidis et al., 2008), constituting thus a small-scale coupled sea-atmosphere system with a short time response to climatic forcing (Xoplaki et al., 2004). The Mediterranean climate is also influenced by

weather conditions over the Atlantic and sometimes by polar outbreaks. However, the key factor of the Mediterranean climate is its seasonality, marked by a strong annual precipitation cycle between dry and wet seasons (Dünkeloh and Jacobeit, 2003; Lionello et al., 2006).

Since the majority of Mediterranean ecosystems are water limited and depend on the seasonal and temporal dynamics of precipitation (Blondel et al., 2010), the Mediterranean forests are highly vulnerable to future climate changes. It is expected

that that by 2100 the annual rainfall will drop by up to 20 % (up to 50 % less in summer), and the mean temperatures will increase by 3-4 °C (Solomon et al., 2007; Giorgi and Lionello, 2008). During the last decades, the intensity and frequency of drought and fire events have increased in the Mediterranean region (Solomon et al., 2007). Even more extreme droughts and warmings have been reported in the palaeoclimatological data (Carrión et al., 2010; Tarroso et al., 2016). Precipitation has been a key climatic variable both in the history of vegetation and in the demographic and cultural dynamics of the

Mediterranean Basin, and particularly in the Iberian Peninsula (Ninyerola et al., 2007; Benito-Garzón et al., 2008; Pontevedra et al., 2017), where the combination of archaeological and palaeoenvironmental studies has shown the influence of abrupt climatic events on settlement patterns and selective ways of anthropogenic exploitation of ecosystems (Carrión et al., 2010; Lillios et al., 2016; Blanco-González et al., 2018).

Given the steep gradients and the coupling between vegetation and water availability, past vegetation changes in the Iberian

Peninsula provide a means to investigate past water availability and precipitation changes. In the last decades, a number of studies based on lake level, pollen, and speleothem data have dealt with synthetic climate reconstructions in the Iberian Peninsula (Tarroso et al., 2016; Morellón et al., 2018). Here, we report pollen-based quantitative precipitation reconstruction results based on a transfer function approach from seven pollen records from different parts of the Iberian Peninsula following a North-South transect from the Atlantic to the Mediterranean climatic domain. To provide a regional synthesis of

the precipitation and humidity changes, we compare our pollen-based precipitation reconstructions with independent records of humidity, such as lake-level data from the Iberian Peninsula, from ~15,000 calibrated years before present (cal yr BP) to the present. This time period is interesting because it encompasses the Pleistocene-Holocene transition, a major transition in climate and vegetation history, and the rapid Late Pleistocene climatic changes documented in the ice core records and in the records from the European continent.

Orbitally-induced differences in seasonal insolation have mainly determined the long-term Holocene climatic evolution in Europe, involving a fairly distinct thermal maximum during the early and mid-Holocene in the high latitudes, followed by a transition to colder conditions around 5000 cal yr BP (Renssen et al. 2009; Marcott et al. 2013). For southern Europe, contrasting interpretations especially about the mid-Holocene climatic conditions have been presented on the basis of biotic proxy data (Mauri et al. 2015; Samartin et al. 2017). Attempts to unravel climate variability in the Holocene in the



Mediterranean region are complicated by interactions between human activity and natural environmental changes, especially those that have occurred from the mid-Holocene to the present day (Carrión et al., 2000). Therefore, multiproxy studies and detailed high-resolution palaeoclimatic reconstructions are required to disentangle the various climatic signals merged in palaeorecords. Pollen based climatic reconstructions in the Iberian Peninsula are still fragmentary, but they are important and

interesting because they provide a range of quantitative precipitation estimates that are understandable in comparison with present climate, allow the testing of predicted climate changes under scenarios of future climate change, and help understand their effects on flora and fauna (Pérez-Díaz et al., 2017). Moreover, they are directly associated with archaeological and cultural trends and events, and they contribute to the prehistoric development of human societies under changing climatic and environmental conditions in the Mediterranean basin.

**2.     Material and methods**

**2.1.     Study area**

Our study area is in Spain, following a North-South transect. It extends over a surface of 505,990 km2 from latitude 43° 47´N to 36° 01´ N and longitude 9° 30´ W to 3° 19´E, encompassing several mountain ranges. The mean elevation of the Iberian Peninsula is around 660 m.

The climate of the peninsula is divided into two major climate zones: (i) the Atlantic climate characterized by mild summers and cold, rainy winters; and (ii) the Mediterranean climate with mild winters and hot, dry summers (Capel, 2000). The Atlantic Ocean influences the northern and western parts of the peninsula, and the Mediterranean Sea influences the South (Fig. 1). The coastline is under the influence of the Atlantic Ocean in the North and the West, and the Mediterranean Sea in the South. The lowest temperatures are measured in the regions influenced by the Atlantic Ocean and the highest in the

regions adjacent to the Mediterranean Sea and the Sahara desert. In addition to a general North-South climatic gradient, seasonal and diurnal thermal gradients stretch from the coast to the centre of the peninsula (Dasari et al., 2014). From a biogeographical point of view, the Eurosiberian bioregion extends from Galicia, northern Portugal, Asturias, Cantabria, the Basque Country and the western and central Pyrenees. It is characterized by a wet climate, moderated by the oceanic influence, with temperate-cold winters and no clearly defined dry season. The Mediterranean bioregion incorporates all

inland plateaus and mountains as well as the Mediterranean basin zones (Rivas-Martínez, 2007). The main vegetation types vary from semi-desertic flora, Mediterranean oak forests, steppeland areas and evergreen pine forests, to deciduous and high-mountain pine forests, and subalpine and alpine vegetation (Blanco-Castro et al., 1997).

**2.2.     Data sources**

The use of transfer functions for quantitative climate reconstructions requires a collection of modern pollen samples that can

be used as a training set in the reconstruction. Our modern pollen-climate training set includes 236 modern pollen samples with known modern annual precipitation (Pann) values analysed for relative abundances of 136 pollen taxa (Fig. 1). Modern



pollen surface samples (moss polsters) were collected with positional and altitudinal data recorded using a portable GPS device, following North-South and East-West transects in Spain (Fig. 1). Several moss samples were randomly collected on the ground at each site within an area of 100 m$^2$ and homogenized into one sample. The collection approach ensured a representative sampling of flora with either long-range or short-range pollen dispersal and also minimized local

overrepresentation of single species. Sites were chosen using the Vegetation Map of Spain (Rivas-Martínez, 2007) to properly characterize the major vegetation communities. The samples were treated with the standard techniques (Moore et al., 1991). A minimum of 500 pollen of taxa belonging to pollen sum taxa were counted while aquatic taxa were excluded from the pollen sum. To establish criteria of standardization and consistency in the data and to reduce bias, only taxa with percentages > 1 % and present in at least 5 % of the samples were included. Following this procedure, 136 pollen taxa were

selected and the percentages were recalculated accordingly. Modern annual precipitation values were obtained from the WorldClim database (Fick and Hijmans, 2017) in a 30-sec resolution (approximately 1 km$^2$). Annual precipitation values for the surface sites range from 231 mm to 1327 mm with gradient 1096 mm. For more details on the modern pollen samples see López-Sáez et al. (2010, 2013, 2015) and Davis et al. (2013).

The pollen records on which the past precipitation reconstructions are based are peat cores from seven bogs: Alto de la

Espina (or La Molina), El Maíllo, Monte Areo, Navarrés-3, Quintanar de la Sierra, San Rafael and Zalama (Fig 1). They were gathered from the European Pollen Database (http://www.europeanpollendatabase.net), the Spanish research project Paleodiversitas (Carrión, 2015), or directly provided by researchers (Table 1). Pollen percentages (Fig. S1) were calculated from terrestrial pollen sums, excluding ferns and aquatic plants. Sites Alto de la Espina, Monte Areo and Zalama are located in the Eurosiberian region, while El Maíllo, Navarrés-3, Quintanar de la Sierra and San Rafael are located in the

Mediterranean region (Fig. 1). In the Eurosiberian region, Zalama is located at the highest altitude, followed by Alto de la Espina and Monte Areo. Quintanar de la Sierra belongs biogeographically to the Mediterranean region, but is located in the heart of the northern Iberian Range that can be considered as an "island of Eurosiberian vegetation". El Maíllo is located in valley area of the peninsular centre and Navarrés-3 is in an area close to the coast of the Mediterranean Sea. San Rafael is in the most southeastern zone.

The chronologies of all sites are based on radiocarbon dating. To produce chronology for each fossil pollen record we used Bayesian age-depth model called Bchron (Haslett and Parnell, 2008). Bchron first calibrates radiocarbon dates with a calibration curve (IntCal13) and then fits the age-depth model, which is consistent with the calibrated radiocarbon dates. Assumptions for the age-depth model are continuous, monotone and piecewise linear age-depth dependence. The age for the uppermost sediment of the core was assumed to be the year when the core was extracted. Fig. 2 shows the results of the

seven Bchron runs. The figure shows the posterior distributions of the calibrated radiocarbon dates, the posterior mean chronology, and the 95 % credible intervals for the possible chronologies. For Alto de la Espina, Monte Areo, El Maíllo and Zalama the Bchron chronologies seem to be reliable for the whole core. For Quintanar de la Sierra we include only the last 14,500 years in the reconstruction since before this date the chronology becomes too uncertain because the radiocarbon dates are remarkably inconsistent. For San Rafael five AMS dates suggest a fairly stable and reliable Holocene sedimentation rate.





However, the Late Pleistocene sequence is based only on one date, and we consider therefore the Late Pleistocene sequence's chronology poorly constrained, and exclude it from the palaeoclimate reconstructions. The record from Navarrés-3 begins 12,000 cal yr BP because before that time the chronology becomes unreliable in the lower parts of the core. The core, however, ends about 3000 cal yr BP, thus missing the late Holocene part, making the Navarrés-3 a chronologically

floating sequence (Fig. 2). All ages in the text are expressed as cal yr BP.

## 2.3. Reconstruction of past climate variables

The selection of the climate variable of interest is a critical step in quantitative climate reconstructions (Li et al., 2015). In the Iberian Peninsula, and in larger context in the whole Mediterranean region, where summers are hot and dry, water availability is generally considered the critically important climatic variable for plant populations and communities, and its

regional and temporal changes greatly influence the vegetation structure and composition (Vicente-Serrano et al., 2014; Samartin et al., 2017; Vidal-Macua et al., 2017). However, the summer temperature may also be an important factor especially at the high altitudes (Vidal-Macua et al., 2017). It is realistic to accept that no single climatic variable can account for the complete influence of climate on vegetation and that no single or few reconstructed climate variables can capture the full spectrum climate patterns and changes in the past. Given that our seven pollen records are from sites located at altitude

lower than 1500 m a.s.l., the climate variable we have reconstructed is annual mean precipitation (Pann). In our study region, Pann is an ecologically important and conceptually simple variable, which can be used in comparisons with other palaeoclimate records and model simulations. Precipitation has a clear zonal pattern in the Iberian Peninsula, and its importance for vegetation patterns is reflected by the comparable zonality of vegetation. In the leave-one-out cross-validation test, Pann has high $r^2$ and low RMSEP (Table 2), demonstrating that it accounts for a large proportion of variance

in the precipitation-related climatic patterns in the region.

We use two different, complementary quantitative techniques, weighted-averaging partial least squares regression technique (WA-PLS) and Bayesian modelling, to test the performance of our modern pollen-climate training set and to produce the past precipitation reconstructions from the seven pollen records. With both techniques, all 236 modern pollen samples were used for constructing the transfer functions for modern annual precipitation (Pann). WA-PLS is a non-linear, unimodal

regression and calibration technique commonly used in quantitative environmental reconstructions (Juggins and Birks, 2012). In all cases, we used a two-component WA-PLS model by ter Braak and Juggins (1993). Training set pollen data values (as percentages) were square root transformed for WA-PLS regression in order to reduce noise in the data. Calculation of WA-PLS transfer functions was performed in the C2 programme (Juggins, 2007).

The Bayesian reconstruction method used is based on Bummer, a Bayesian hierarchical multinomial regression model

introduced in Vasko et al. (2002). In the basic Bummer model, the observed pollen taxon relative abundances are modelled by a multinomial distribution, where the taxon occurrence probabilities are treated as Dirichlet-distributed random variables whose distribution is determined by the pollen environmental response parameters as well as the mean annual precipitation. The taxon environmental response is modelled by a unimodal Gaussian function, with shape and mean determined by the



response parameters alpha (scale), beta (optimal precipitation) and gamma (tolerance); See Figure S2. The prior distributions of the model parameters are listed in Table S1.

The performance of both transfer functions was evaluated by leave-one-out cross-validation (Birks et al. 1990). Based on the leave-one-out cross-validation results we calculated the coefficient of determination ($r^2$), root-mean-square error of prediction (RMSEP) and maximum bias as performance statistics.

## 3.        Results and Discussion

### 3.1. Transfer function performance

Leave-one-out cross-validation performance statistics ($r^2$, RMSEP and maximum bias) for the two-component WA-PLS and Bayesian transfer functions are shown in Fig. 3 and Table 2. The $r^2$ between the observed modern values and those predicted by WA-PLS in cross-correlation test is 0.61 and in the Bayesian model 0.55 and the RMSEP is 145 mm with WA-PLS and 170 mm with Bayesian model. Thus WA-PLS slightly outperforms the Bayesian model as measured with RMSEP, $r^2$ and maximum bias. One potential reason for this is that in the Bayesian model we used a wide priori for the predicted precipitation in order not to restrict the precipitation values too much a priori. When these performance statistics are compared with other validation tests with WA-PLS and Bayesian-based transfer functions, it can be seen that they are reasonably high, but still slightly lower than in other regional models. For example, in northern Europe, $r^2$ values between the predicted and observed summer or annual mean temperature values are generally 0.7 to 0.85 (Seppä and Bennett, 2003; Birks and Seppä, 2004), in China for Pann 0.8 (Li et al., 2016) and in training set from the Swiss Alps as high as 0.9 for mean summer temperature (Lotter et al., 2000).

There are a number of reasons, which can explain the slightly lower performance statistics of the pollen-climate calibration set in the Iberian Peninsula as compared to northern Europe. One undeniable factor is the long-lasting and intense human impact that causes bias in the climate-vegetation relationships (Carrión et al., 2000; López-Sáez et al., 2016) and blurs the performance of the pollen-climate transfer functions (Li et al. 2015). Another likely source is that the fossil pollen samples and modern samples in the training set represent different sedimentary environments. Besides having consistent taxonomy and nomenclature and being of comparable quality, the modern pollen data should be from the same sedimentary environment (e.g., lakes of similar size) as the fossil data-sets used for reconstruction purposes (Seppä et al., 2004; Birks et al., 2010). Unfortunately it is not possible to use pollen samples from the same sedimentary environment for the training set and fossil data in the Iberian Peninsula. The fossil assemblages are from mires, but the modern samples in the training set represent locally integrated moss samples. This is a common problem when constructing pollen-climate transfer functions in dry and semidry regions, with a limited number of lakes and peat bogs (Pontevedra et al., 2017).



## 3.2. Evaluation of the reconstructions

The results of the Pann reconstructions are shown in Figs. 4 and 5. The shapes of the reconstructions based on WA-PLS and Bayesian modelling are comparable but there are some differences in the levels of reconstructed Pann values. In general, the variability is higher in the WA-PLS-based reconstructions, as can be seen especially in the records from Monte Areo, San Rafael and El Maíllo, while the absolute reconstructed Pann values are similar in both reconstruction approaches. It is important to keep in mind that with the Bayesian reconstructions we show only the posterior mean value, which is just one possibility to summarize the Bayesian reconstruction, and therefore comparison to WA-PLS reconstructions is not straightforward. Furthermore, the individual Bayesian precipitation reconstructions have higher variability compared to the posterior mean. When the seven Pann reconstructions are compared, they indicate relatively consistently the main trends (Figs. 4-5). For exploring the generality of our results, we compare them with selected Late Pleistocene and Holocene lake-level that reflect general humidity in the Iberian Peninsula. Additionally, to gain insights to the underlying climatic mechanisms and climatic teleconnections that can explain the reconstructed features, we compare the results with chironomid-based summer temperature records and temperature-related proxy records from the Greenland ice cores (Figs. 6-7), which represent the general Late Pleistocene and Holocene climatic conditions in the northern Atlantic region.

## 3.3 Precipitation trends

### 3.3.1. Late Pleistocene (~14,500-11,600 cal yr BP)

In our dataset, the Late Pleistocene Pann record is only available from the Quintanar de la Sierra pollen sequence, as it reaches back to 14,500 cal yr BP, with a reasonably high resolution (Figs. 4-5). The reconstructed Pann values in the WA-PLS-based record show an increasing trend between 14,500 and 14,250 cal yr BP, rising from 700 to 900 mm; a prolonged decrease until 13,900 cal yr BP (< 800 mm), and, finally, an oscillating curve with relatively constant values of about 800-850 mm until 12,900 cal yr BP (Fig. 4). A similar tendency is observed in the Bayesian reconstruction although with slightly higher values, reaching a maximum higher than 1000 mm ~14,250 cal yr BP (Fig. 5). These features are generally consistent with the main climatic trends in Europe during the Late Pleistocene. The period with higher Pann from 14,500 to 12,900 cal yr BP corresponds with the Greenland interstadial 1 (GI-1), or Bølling-Allerød interstadial, with higher temperatures in northern Europe, and the subsequent period 12,900-11,700 cal yr BP corresponds with the Greenland stadial 1 (GS-1), or Younger Dryas stadial, clearly reflected in the Greenland ice core data (Fig. 6). The correspondence between the higher precipitation with higher temperatures and lower precipitation with lower temperatures in Europe can been also seen in the comparison with chironomid-based summary temperature curve for the Late Pleistocene (Fig. 6).

In a more precise comparison with Late Pleistocene records from the Iberian Peninsula, our results are in agreement with those presented by Naughton et al. (2016) and Tarroso et al. (2016), who also describe the GI-1interstadial as a period of increase in humidity with a relatively stable rainfall pattern in northern Iberian Peninsula, which in the case of Quintanar de la Sierra is evident only from 13,900 cal yr BP. These climatic conditions of greater humidity allowed the survival of the



beech (*Fagus sylvatica*) in late-glacial refugia of the northern Iberian System, particularly during the first half of the Bølling-Allerød interstadial (López-Merino et al., 2008). Although the presence of beech has not been confirmed in this period in Quintanar de la Sierra (Peñalba, 1994; Fig. S1), it is documented in other neighbouring pollen records such as Grande Lake (Ruiz-Zapata et al., 2003). Beech forests are considered typical elements of the Eurosiberian region (Rivas-Martínez, 2007),

but some Mediterranean areas of the Iberian Peninsula, i.e. the northern Iberian Range, can be considered as islands of Eurosiberian vegetation. Today, beech forests are only residual in these mountains (López-Merino et al., 2008). Our reconstructed Pann values are also in agreement with other Iberian pollen records showing a clear expansion of deciduous oak forests during the GI-1 interstadial indicating wetter and warmer conditions, summarized by Carrión et al. (2010) and González-Sampériz et al. (2010).

The GS-1 stadial can be seen in the Quintanar de la Sierra reconstruction by a decline of precipitation to 750-650 mm from 12,900 to 11,700 cal yr BP (Figs. 4-5). The Iberian pollen records show usually an increase of *Betula*, heliophilous herbs, *Poaceae*, and shrubland and semi-arid pollen taxa such as *Artemisia*, *Chenopodiaceae*, *Ephedra* during the GS-1, as well as *Pinus* in mountain environments, confirming the aridity during this period. Steppe vegetation has been described during the GS-1 in most Iberian territories (e.g., Allen et al., 1996; Peñalba et al., 1997; van der Knaap and van Leewen, 1997; Ruiz-

Zapata et al., 2003; González-Sampériz et al., 2006, 2010, 2017; Carrión et al., 2010; Fletcher et al., 2010b; Moreno et al., 2011; López-Merino et al., 2012; Muñoz-Sobrino et al., 2013; Iriarte-Chiapusso et al., 2016). By contrast, records from the inland Mediterranean environments and southeastern region reveal little changes in vegetation and suggest the persistence of conifers and open landscapes during the GS-1 (Carrión and van Geel, 1999; Vegas et al., 2010; Aranbarri et al., 2014).

The Quintanar de la Sierra record also concurs with the lake-level reconstructions for the GS-1 (Morellón et al., 2018; Fig.

7). In particular, Estanya Lake, at lower altitude in the Pyrenees, reflects large climatic change during the GS-1, with the onset of a marked decrease in the lake level, increased salinity, and a sudden decline in organic productivity with the absence of diatoms (Morellón et al., 2009). The Enol Lake record also shows very low productivity and low carbonate content during the GS-1, which can be interpreted as an indication of lower temperatures and a decline in precipitation (Moreno et al., 2011). A similar scenario was reconstructed in Roya lagoon in northwestern Iberia, where a decrease in organic productivity has been interpreted as an indication of colder and drier conditions between 12,700 and 11,700 cal yr BP (Muñoz-Sobrino, et

al., 2013). The Grande Lake also shows a marked dry period from 12,600 cal yr BP until a rise at 11,700 cal yr BP, as reflected by the deposition of rhythmites and a shift in diatom and pollen assemblages (Ruiz-Zapata et al., 2003; Vegas et al., 2003). Similar to this, records from the easternmost sites located near to the Mediterranean coastal areas recorded markedly arid conditions during most of the GS-1 either in two phases or in a more continuous pattern (Morellón et al., 2018). There is

thus substantial multi-proxy evidence suggesting that the Pann was relatively high during the period 14,500 to 12,900 cal yr BP (GI-1), and lower during the period 12,900 to 11,700 cal yr BP (GS-1).



### 3.3.2.    Early Holocene (~11,600-8200 cal yr BP)

The early Holocene in the Quintanar de la Sierra record is characterized by a marked peak in the Pann values, up to over
1000 mm at 11,600 cal yr BP, followed by a progressive decline to under 800 mm until 8200 cal yr BP (Fig. 4). The other
records covering the early Holocene show similar features with a period of maximum Pann values at ~11,600-11,000 cal yr

BP and a later period with progressively decreasing values until 8200 cal yr BP (Figs. 4-5). These patterns are especially
evident in the records from the Eurosiberian biogeographic region (e.g., Monte Areo, Alto de la Espina), while those in the
Mediterranean region show more gradual and irregular trends (San Rafael, Navarrés-3, El Maíllo).

It is notable that in our reconstructions, the rise of the Pann values in the early Holocene around 11,000 cal yr BP appears
synchronous between the Eurosiberian and Mediterrean regions, as can be seen in the reconstructed values in the Quintanar

de la Sierra, San Rafael, and Navarrés-3 records (Figs. 4-5, 7). This contrasts the earlier interpretations that the development
of the vegetation in the Iberian Mediterranean region was quite different than in the Eurosiberian region, as the persistence of
conifer populations continued during the early Holocene, showing only minor oscillations in the mesophilous pollen
frequencies in respect to the preceding GS-1 (Carrión et al., 2010). However, it is also important to note that the Pann trend
suggested by our data, with a higher Pann values in the early Holocene, with Quintanar de la Sierra, Monte Areo and San

Rafael records reaching the maximum Holocene Pann values as early as about 11,600 cal yr BP, followed by a progressively
lower values until 8200 cal yr BP is not fully compatible with the lake-level reconstruction data. For example, in the
reconstruction from Estanya Lake, located in the transitional area between the humid Pyrenees and the semi-arid Central
Ebro Basin in northeastern Spain, the onset of the Holocene (~11,600-9400 cal yr BP) is characterized by low lake levels
(Fig. 7), with a shallow, ephemeral, saline lake-mud flat complex with carbonate-dominated sedimentation during the

flooding episodes, and gypsum precipitation during desiccation phases (Morellón et al., 2009), which also affected the
development and preservation of diatom communities. These differences between different types of proxy records show that
the Early-Holocene precipitation and moisture conditions in the Iberian Peninsula are still poorly understood and that more
high-resolution reconstructions are needed to solve the inconsistencies between different types of data and the outstanding
questions.

### 3.3.3.    8.2 ka event

The clearest short-lived abrupt event in the Holocene records in the North Atlantic-North European region is the  8200 cal yr
BP (8.2 ka) cold event (Alley et al., 1997). This event has been detected in many pollen records from the Iberian Peninsula
(López-Sáez et al., 2008). On the Mediterranean coast and in the middle Ebro valley, it is characterized by the progression of
Mediterranean pine and evegreen oak forests and the decline of deciduous oak (Davis and Stevenson, 2007, while in the

eastern territories (e.g., Les Alcusses and Navarrés; Fig. S1) the high-mountain pine forests more adapted to a cold
continental climate expanded, while the Mediterranean vegetation in lower and inner areas was reduced (Carrión and van
Geel, 1999; Tallón et al., 2014). In the semi-arid region in the southeast, the San Rafael pollen record points out the



development of grasslands and xerophytic vegetation (Pantaleón-Cano et al., 2003; Fig. S1). Changes in lake level also indicate increased aridity, with desiccation during this period at Medina Lake in the southwest (Reed et al., 2001) and at Villafáfila lakes in inland Iberia (López-Sáez et al., 2017). In our results, in the Alto de la Espina record, the reconstructed Pann drops to under 800 mm, but this takes place 8000-7900 cal yr BP, and in the Quintanar de la Sierra and San Rafael

records there is a dip between 8300-8100 cal yr BP, but it is indicated only by one data point (Figs. 4-5). Thus there is no unequivocal evidence for this event in our data in the Mediterranean or Eurosiberian regions, but we cannot exclude its possibility either, and conclude that accurately dated high-resolution pollen records are needed to firmly detect the nature of the 8.2 ka event in quantitative Pann reconstructions.

### 3.3.4. Mid Holocene (~8200-4200 cal yr BP)

The Mid Holocene from 8200 to 4200 cal yr BP is characterized by higher Pann values in most of the reconstructions (Figs. 4-5), exceeding 1000 mm in Alto de la Espina and Monte Areo within the Eurosiberian region, and 900 mm in Quintanar de la Sierra. San Rafael and Navarrés-3, the two records located in the Mediterranean region, show clearly higher Pann values throughout the study period, with mean values between 400-600 mm. The Zalama and El Maíllo records are clear exceptions from this general trend, as in these records the Pann values remain constant around 900 mm. However, in most cases, in the

reconstructed Pann values are above the Holocene means and the mid-Holocene is thus the longest and most prominent humid period reflected in our records.

The Mid-Holocene humid period can be also generally observed in other records from the Iberian Peninsula. The reconstructions from Basa de la Mora Lake in northeastern Spain shows a period of highest Holocene lake levels from 8100 to 5700 cal yr BP (Pérez-Sanz et al., 2013; González-Sampériz et al., 2017) and the Estanya lake-level reconstruction based

on sedimentary facies analysis suggest a period of high lake levels from 8200 to 4200 cal yr BP, supported by a period of low but variable water salinity (Fig. 7) (Morellón et al., 2009, 2018). In the southern Iberian Peninsula, the period with highest humidity has been dated to 9500-7600 cal yr BP (Ramos-Román et al. 2018a). In addition, palaeoclimate reconstructions from central Pyrenees, based on chironomids, and thus independent of pollen data, indicate that the summer temperatures were highest from 8800 to 6200 cal yr BP (Tarrats et al., 2018). Thus the period from roughly 8000 to 5000 cal

yr BP was characterized by high summer temperatures and higher than present precipitation in our study region.

In larger regional context this mid-Holocene period with high Pann corresponds with the Holocene thermal maximum (HTM) in the high latitudes (Renssen et al., 2009). In the Mediterranean region, including the Iberian Peninsula, the climate of this period has been long debated. In many pollen-based climate reconstructions the period from 8000 to 5000 cal yr BP has been seen as a period of cool summers (Davis et al., 2003; Mauri et al., 2015), which contradicts with the output of

climate models for the period. Samartin et al. (2017) used chironomid-based summer temperature reconstructions from Italy to argue that the summer temperature during the period has been higher than at present, in line with the models, and postulated that the pollen-based reconstructions of Mid-Holocene temperatures in the Mediterranean region are biased by the human influence. The inferred warm summers in the chironomid-based reconstruction from Basa de la Mora Lake in the



Pyrenees support this argument (Tarrats et al. 2018). We did not aim to use pollen data for summer temperature reconstructions for the reasons explained earlier, and our results do not contribute directly to this debate, but in general we agree with these authors in that the predominant driver of vegetation patterns in the Mediterranean region is water availability and not summer temperature, and this is a fact which must be borne in mind when assessing any feature in
pollen-based temperature reconstructions in this region.

### 3.3.5.   Late Holocene (~4200 cal yr BP-present)

Our records suggest a declining general trend of Pann over the last 5000 years (Figs. 4-5). This is clearest in the records from Quintanar de la Sierra, Alto de la Espina and San Rafael, where the Pann values decline to about 500-200 mm from the mid-Holocene Pann maximum. When the modern Pann values are compared with the maximum values at 8000-5000 cal yr BP, a
50 % reduction of Pann in Spain is indicated. The record from Zalama differs from this trend, as it shows a fairly stable Pann trend over this period. In the Alto de la Espina record, a short-lived peak of anomalously high Pann values is indicated at 1500 cal yr BP. As shown in the pollen diagram, these values are caused by the exceptionally high *Pteridium* spore values, reaching a maximum up to 83 % at 1500 cal yr BP (Fig. S1). Such a peak of *Pteridium* is clearly an anomaly, probably caused by a local over-representation of *Pteridium* population at the coring site on the Alto de la Espina bog.
The increasing dryness over the last 5000 years evident in our Pann records (Figs. 4-5) has been observed in many records from the Iberian Peninsula. In the Basa de la Mora record, the lake level falls from 6000 cal yr BP to 4000 cal yr BP, with the period of lowest Holocene level from 3500-2300 cal yr BP, followed by a slight rise over the last two millennia (González-Sampériz et al., 2017). In Estanya Lake record, the more saline and shallower conditions are seen between 4800 and 1200 cal yr BP, as indicated by the deposition of gypsum-rich sediment and massive sapropels facies (Morellón et al.,
2009). Similarly, the multiproxy data from the Padul record in Sierra Nevada in southern Spain show clear evidence for aridification over the last 4000 years (Ramos-Román et al. 2018b).
A characteristic feature in our Pann reconstructions is a high variability in the records and between the records during the last 2000 years. The drop of reconstructed Pann from 1500 mm to under 600 mm during the last 500 years in the Monte Areo record is an extreme example of this pattern, suggesting that these wiggles do not represent realistic changes in the Pann
values, but reflect more likely noise in the data. One reason for such variability may be the increasing human impact on vegetation in the Iberian Peninsula. The earliest evidence of agriculture in the Iberian Peninsula is documented in the eastern territories ~7500 cal yr BP during the Early Neolithic. Between ~7500-7000 cal yr BP, agriculture spread across the peninsula (Peña-Chocarro et al., 2018).
In general, the pollen-based climate reconstructions in the Eurosiberian and Mediterranean regions are in most cases strongly
influenced by the human impact on vegetation, including cultivation, forestry, husbandry, burning and clear cutting (Carrión et al., 2010; López-Merino et al., 2014; Lillios et al., 2016). Over the centuries, the human influence has caused the original natural vegetation to shift towards semi-anthropogenic ecosystems, creating novel plant communities such as olive, chestnut, walnut and cork-oak woodlands, or promoted disturbance-adapted sclerophyllous vegetation types. The problem of human





influence in the Mediterranean region has been observed in many pollen-based climate reconstructions. For example, in a reconstruction from Accesa Lake in central Italy, the last 2000 years were excluded from the reconstruction because the pollen signal is strongly dominated by human impact (Peyron et al., 2011) and in a record from Lago di Pergusa in southern Italy, the climate reconstructions based on the pollen data were shown to be biased by the decline of Mediterranean tree

pollen and the increase of herb pollen and other anthropogenic indicators over the last 3000 years (Sadori et al., 2013). Such a long-lasting and intense human impact adds another factor changing the vegetation composition, blurring the detection of the climatic signals, and is reflected as high local variability in pollen-based climate reconstructions (Li et al., 2015).

## 4.        Conclusions

Precipitation is a key driver for the ecosystems in the Iberian Peninsula and changes in its amount and spatial and temporal

distributions have an impact on vegetation history, human activities, and natural hazards. It is thus an important variable in climate and palaeoclimate studies. Thus far most of the reconstructions of changes in past precipitation in the Iberian Peninsula have been based on qualitative and indirect data, such as inferred vegetation changes or changes in lake levels. We have constructed a modern pollen-climate calibration set specifically for the Iberian Peninsula and used it to provide quantitative precipitation reconstructions from fossil pollen cores from different climatic regions. The results show that

precipitation in the Iberian Peninsula has had a strong spatial and latitudinal gradient during the last 15 thousand years. The reconstructed Pann values are clearly higher in northern Spain than at the two sites in the Mediterranean region. The Late Pleistocene is characterized by rapid shifts in Pann values, with the dry period during the GS-1 corresponding the low temperatures in northern Europe. The most pronounced period with high Pann values dates to 8000-4000 cal yr BP, and corresponds roughly with high lake levels and high summer temperatures in the Iberian Peninsula. We thus conclude that

this period is comparable with the Holocene thermal maximum in northern Europe. During the Late Holocene the reconstructions are less consistent. One factor explaining this is probably the substantial human impact on vegetation, such as the clearance of forests and the development of cultivated fields, pastures, meadows and heathlands. The pollen-based Late-Holocene climate reconstructions from the Iberian Peninsula are thus substantially biased by the human impact.

**Code and data availability**

Code used to conduct the analysis for the Bayesian model is already published in the journal The Annals of Applied Statistics as Holmström et al. (2015): doi:10.1214/15-AOAS832SUPPC. In order to run the model used in this manuscript the prior distributions need to be set according to Table S1. Data used to create Figures 1, 3, 4 and 5 and Table 2 are available at http://dx.doi.org/10.17632/4pznttrd4h.1 (Ilvonen et al., 2019). Other data are available from the authors upon request.



**Supplement link (will be included by Copernicus)**

**Author contributions**

LI, JALS and HS designed the study. LI performed the simulations and computations required for the Bayesian model and WA-PLS method. JALS, FAS, SPD and JSC provided the data. LI, JALS and HS were mainly responsible for preparing the manuscript, while all authors commented and contributed to the discussion and interpretations

**Competing interests**

The authors declare that they have no conflict of interest.

**Acknowledgements**

This research has been funded by the Academy of Finland (EBOR and GRASS projects), by the European Research Council (project YMPACT), and through the REDISCO-HAR2017-88035-P (Plan Nacional I+D+I, Spanish Ministry of Economy and Competitiveness) and Relicflora-P11-RNM-7033 (Excellence Research Projects Program from the Andalusian Government) projects.

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





**Table 1: Information on the seven fossil pollen records.**

| Site | Latitude N | Longitude W | Altitude (m a.s.l.) | Pann (mm) | References |
|------|-----------|-------------|---------------------|-----------|------------|
| Alto de la Espina | 43º 22' 52'' | 6 º 19' 38'' | 650 | 930 | López-Merino et al. (2011, 2014) |
| El Maíllo | 40 º 32' 48'' | 6 º 12' 35'' | 1100 | 715 | Morales-Molino et al. (2013) |
| Monte Areo | 43 º 31' 44'' | 5 º 46' 08'' | 200 | 881 | López-Merino et al. (2010) |
| Navarrés-3 | 39 º 05'36'' | 0 º 41'00'' | 225 | 429 | Carrión and van Geel (1999) |
| Quintanar de la Sierra | 42 º 01'31'' | 3 º 01' 34'' | 1470 | 743 | Peñalba (1994) |
| San Rafael | 36 º 46' 25'' | 2 º 36' 05'' | 0 | 231 | Pantaleón-Cano et al. (2003) |
| Zalama | 43 º 08'06'' | 3 º 24' 35'' | 1330 | 1059 | Pérez-Díaz et al. (2016) |



**Table 2: Information and performance statistics of the modern pollen-climate training set. Reported statistics based on leave-one-out cross-validation are root mean square error of prediction (RMSEP), coefficient of determination ($r^2$) and maximum bias. The WA-PLS statistics are based on a two-component model.**

| | |
|---|---|
| Number of sites | 236 |
| Precipitation gradient | 231 to 1327 mm |
| Precipitation range | 1096 mm |
| Number of taxa | 136 |
| WA-PLS RMSEP | 144.71 mm |
| WA-PLS $r^2$ | 0.61 |
| WA-PLS maximum bias | 328.08 mm |
| Bayesian model RMSEP | 170.82 mm |
| Bayesian model $r^2$ | 0.55 |
| Bayesian model maximum bias | 239.53 mm |



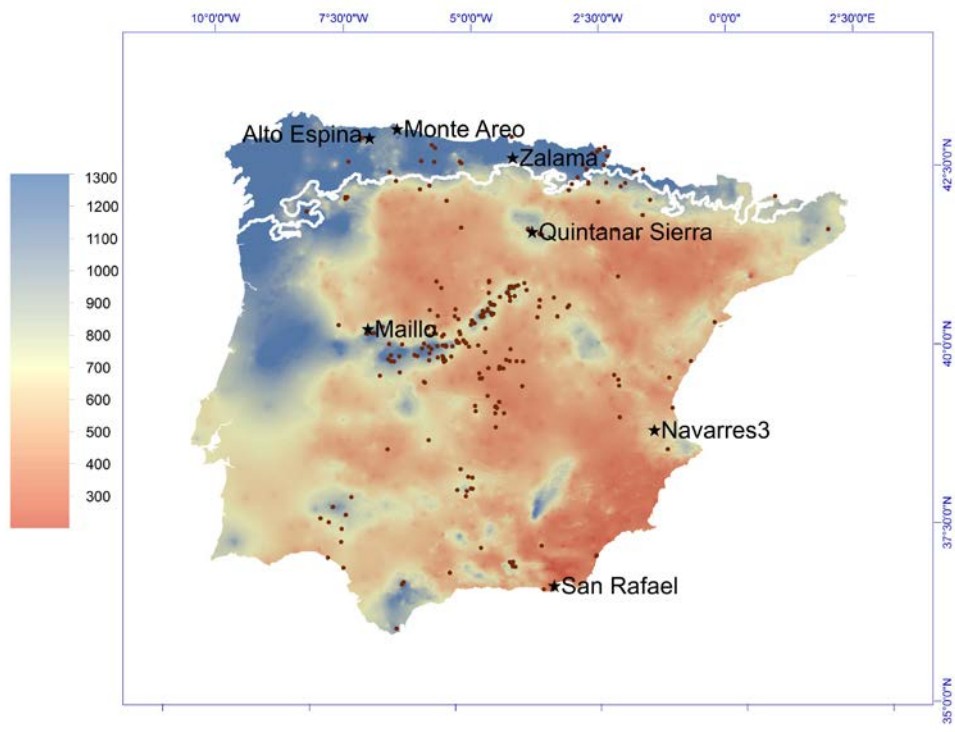

**Figure 1. Locations of training sites and cores. Training sites are denoted with dots and cores are denoted with black stars. The colours in the map indicate the modern Pann (annual precipitation) values in mm yr$^{-1}$.**




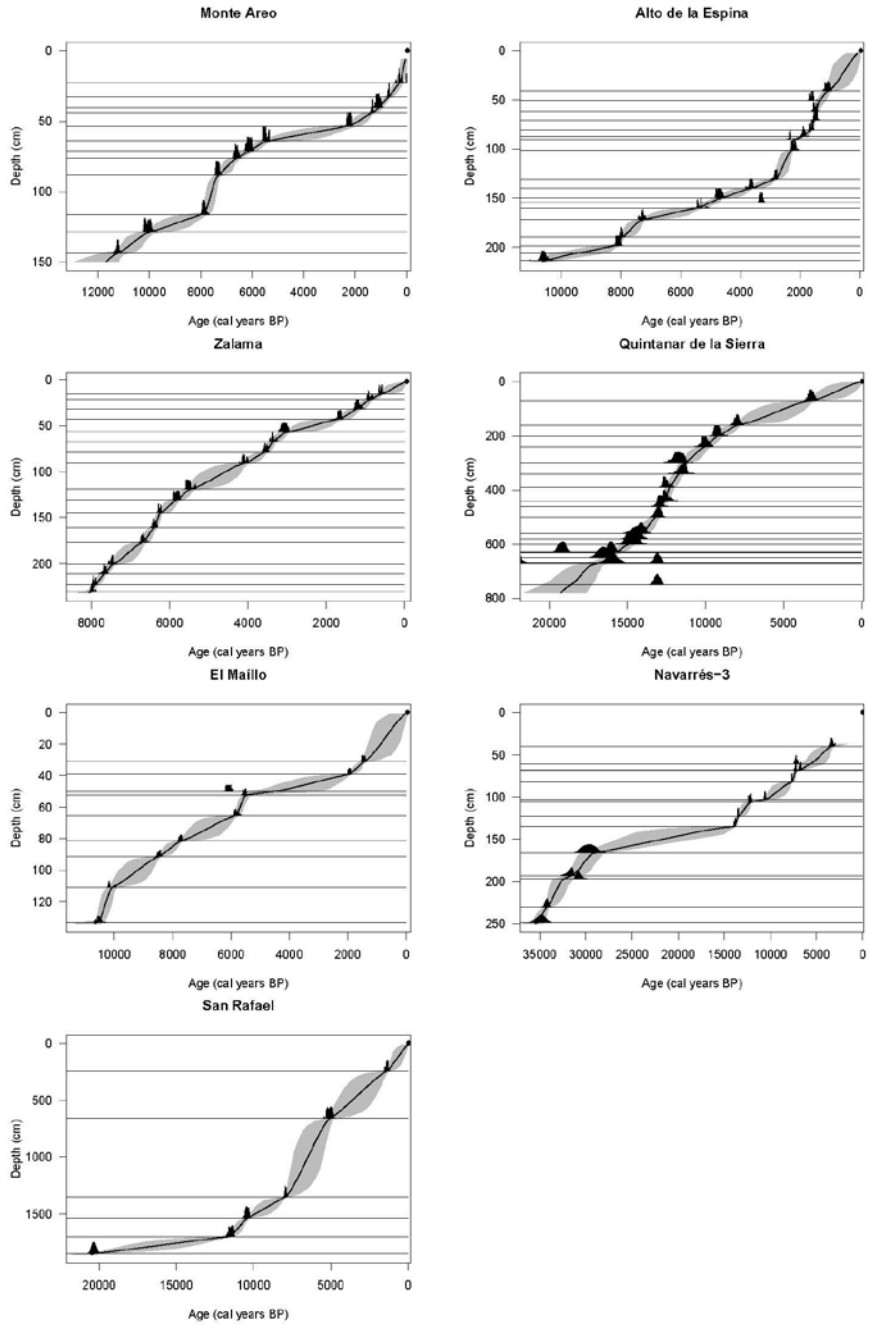

**Figure 2: Outputs of Bchron chronology model run for the seven cores used in precipitation reconstructions. The posterior distributions of the calibrated radiocarbon date are shown in black, the gray lines indicate the radiocarbon dated depths and the 95 % credible intervals for the chronologies are in grey bands. The black line is the posterior mean chronology and the dot marks the top of the core. In the reconstructions we use posterior mean chronologies.**



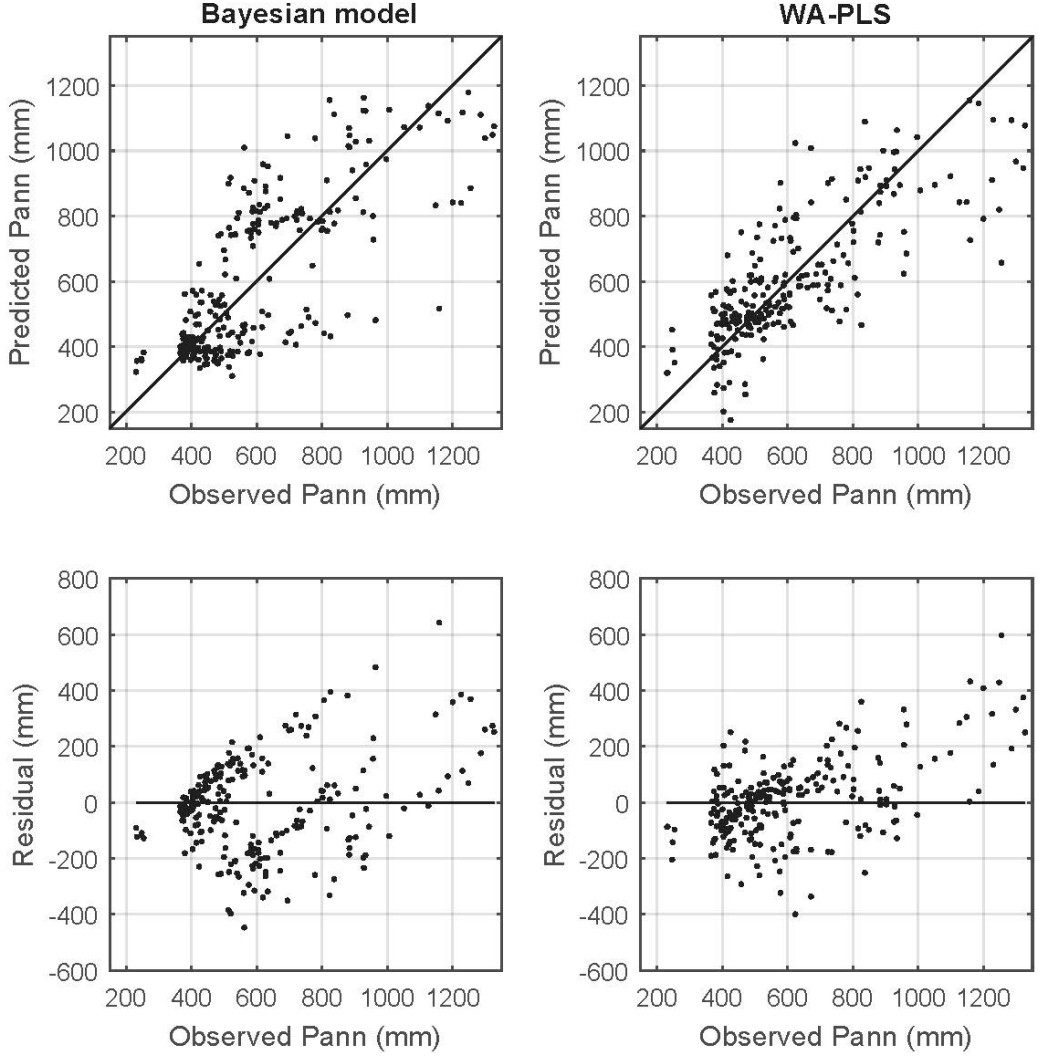

**Figure 3: Plots of predicted versus observed modern Pann and residuals of predicted versus observed modern Pann for Bayesian model and two-component WA-PLS in leave-one-out cross-validation.**



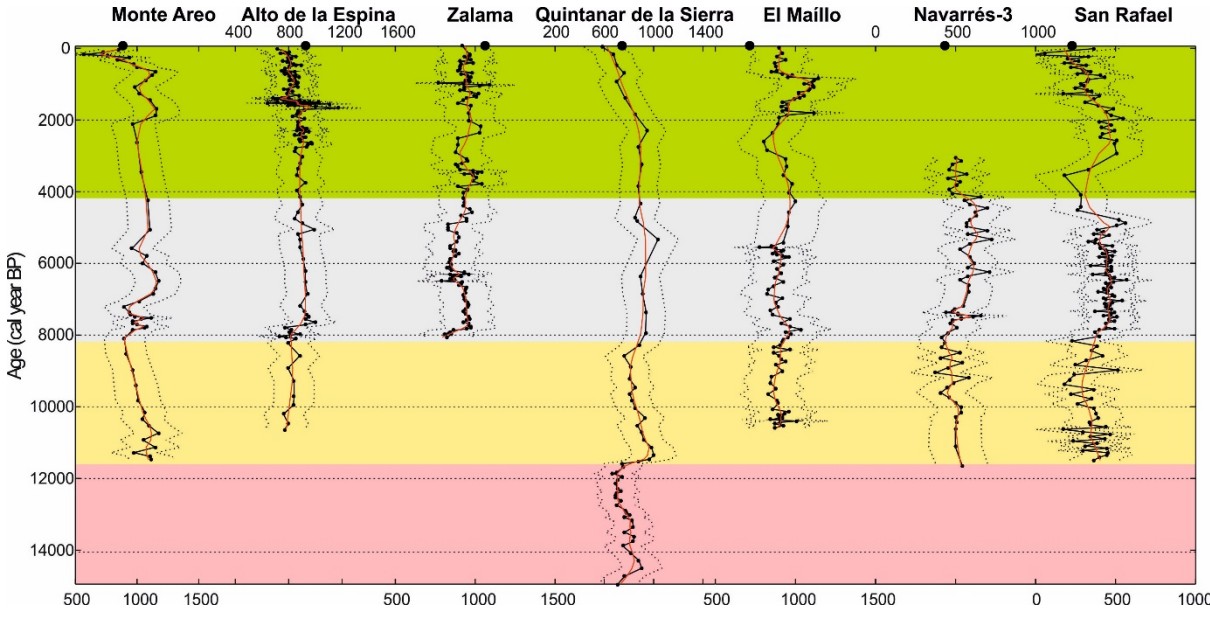

**Figure 4: Pann (mm yr⁻¹) reconstructions for seven pollen records using WA-PLS. The black dots connected with the solid black line are the reconstructed values for Pann, the solid red line is a LOWESS smoother added to the reconstructions (span 0.1) and the black dotted lines denote bootstrap estimated standard errors. The big black dot is the modern value for Pann. X-axis is Pann and y-axis is time in years before present. The colors indicate the formal stratigraphical subdivision of the Holocene and Late Pleistocene.**





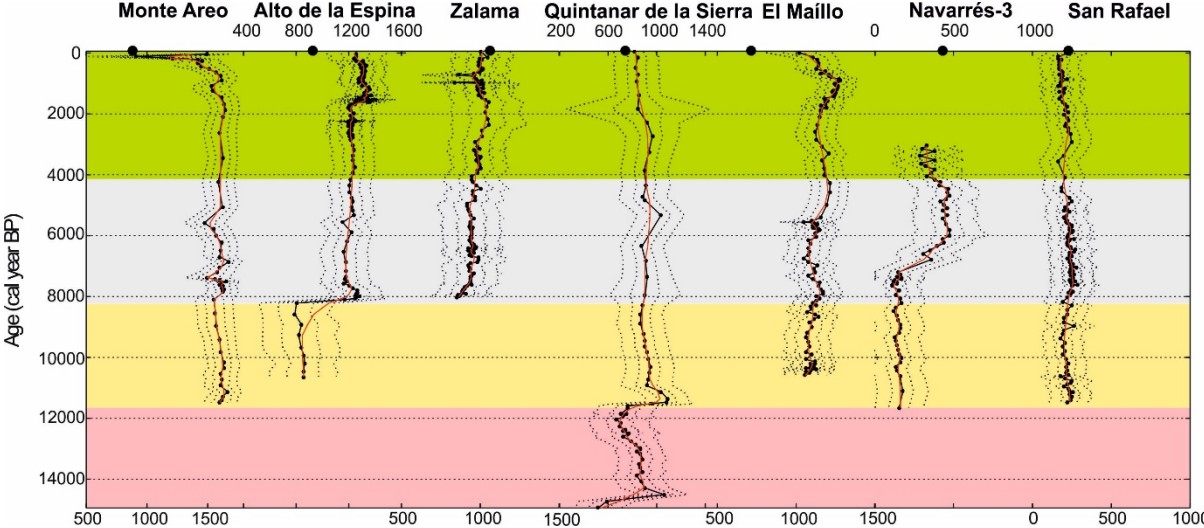

**Figure 5: Pann (mm yr⁻¹) reconstructions for seven pollen records using Bayesian model. The black dots connected with the solid black line are the posterior mean values for Pann, the solid red line is a LOWESS smoother added to the reconstructions (span 0.1). The inner and outer black dotted lines show the point-wise and simultaneous 95 % credible bands, respectively. The big black dot is the modern value for Pann. X-axis is Pann and y-axis is time in years before present.**



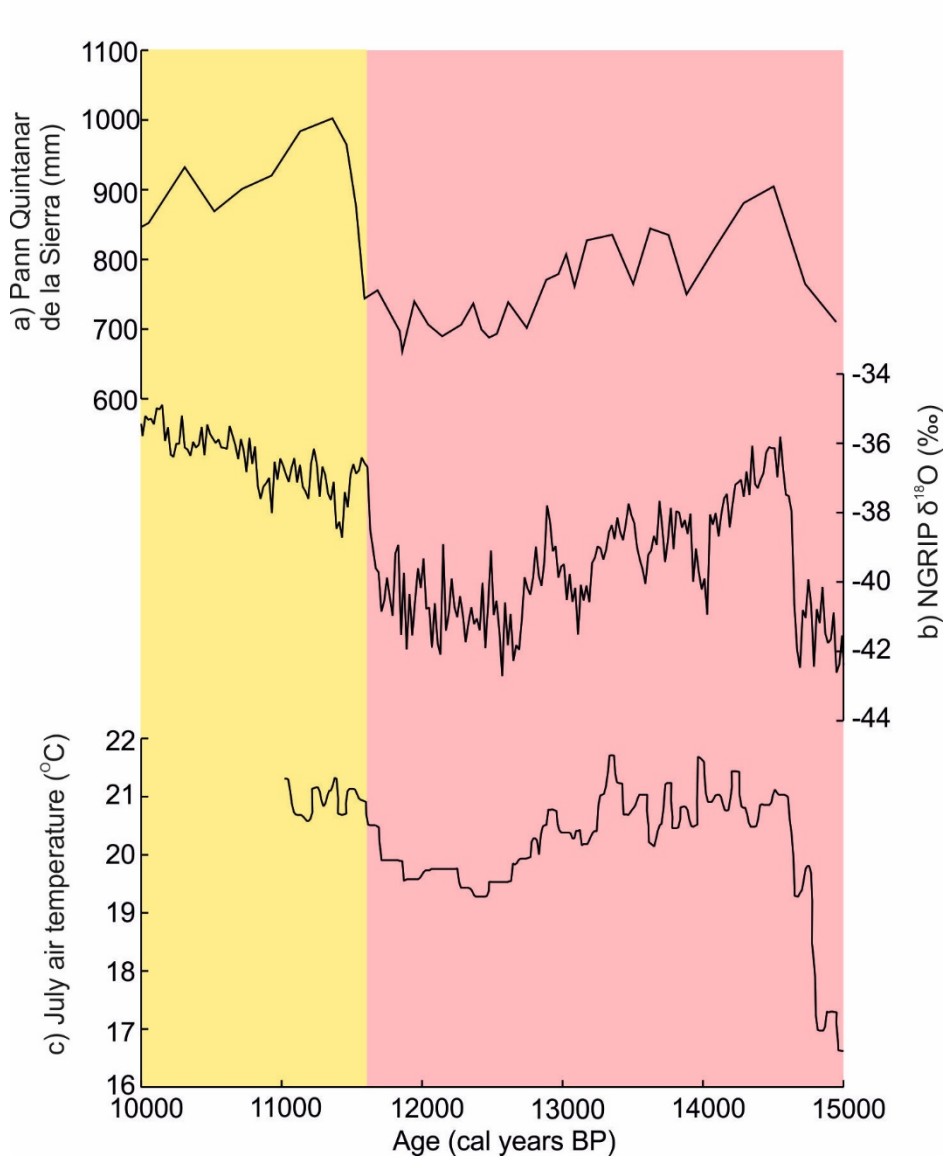

**Figure 6: The Late-Pleistocene precipitation reconstruction from a) Quintanar de la Sierra compared with b) δ¹⁸O record from the NorthGRIP ice core from Greenland (Rasmunssen et al., 2014) and c) chironomid-based July mean temperature reconstruction for SW Europe (Heiri et al., 2014).**



**Figure 7: Reconstructed precipitation trend from a) northern Spain (Quintanar de la Sierra) and b) southern Spain (San Rafael) compared with c) chironomid-based July temperature reconstruction for Basa de la Mora Lake in Central Pyrenees, Spain (Tarrats et al., 2018), d) salinity from lake Estanya. The salinity values (-2, 2) are PCA axis 2 values, with negative values indicating higher salinity (Morellón et al., 2009), e) lake level reconstruction of lake Estanya (Morellón et al., 2009), f) normalized lake level for Laguna Grande (Morellón et al., 2018), g) normalized lake level for Sanabria (Morellón et al., 2018) and h) normalized lake level for Enol (Morellón et al., 2018).**