# Peer review of "Quantitative reconstruction of precipitation changes in the Iberian Peninsula during the Late Pleistocene and the Holocene"

_Climate of the Past, 2019_

## Referee Comment (RC1) · William Fletcher (Referee) · 30 Apr 2019

The discussion paper by Ilvonen et al. entitled "Quantitative reconstruction of precipitation changes in the Iberian Peninsula during the Late Pleistocene and the Holocene" presents new pollen-based precipitation reconstructions from the Iberian Peninsula which are relevant for the wider study of palaeoclimate of the western Mediterranean region. Given the importance of moisture balance for vegetation in the region, this paper addresses an interesting and valuable subject. Also, in light of the strong environmental gradients and climatic diversity of the Iberian Peninsula, it is surprising that there are relatively few studies to date exploring climate reconstruction with this spe-

cific geographical focus. The chosen methods are appropriate for the study with use of two transfer function reconstruction approaches (WA-PLS and Bayesian-based). The paper is concise and clearly presented. The study presents some valuable insights into general trends in reconstructed precipitation levels and engages with current debates in Holocene palaeoclimatology regarding the main climatic signal in Holocene vegetation changes. The approach and findings should be of interest to the community of scientists researching past climatic change in the western Mediterranean region. I would encourage the authors to consider the general observations and comments below, as well as the more specific comments listed, if they seek to enhance the impact of the study in a revised paper for publication in Climate of the Past.

The paper sets out an aim of testing "contrasting interpretations" (p2, line 34) about Holocene climatic conditions, citing the works of Mauri et al., 2015 and Samartin et al., 2017. As far as I understand it, the discrepancy between these cited studies relates to the timing of maximum summer warmth during the Holocene and contrasts between previous pollen- and chironomid- reconstructions for the (western and central) Mediterranean region. I think this discrepancy should be spelled out more clearly in the introduction because some readers may not be familiar with that debate. Going further, the potential implications of overlooking the role of precipitation (in terms of conflating humid- and cool- conditions during the mid-Holocene) could be also made more explicit. Towards the end of the paper, the authors suggest that their findings "do not contribute directly to this debate" – but I suspect that this might not be so clear cut, and the authors might wish to consider that statement (and the implications of their findings) further.

The authors have chosen to focus on precipitation as the target for reconstruction and they present some good reasons why precipitation should be a strong determinant of vegetation cover across the Iberian Peninsula. However, in order to advance the aforementioned debate, it would be helpful to demonstrate that precipitation indeed does a better job than other climatic variables such as summer and winter temperature

at explaining the variance in the modern pollen data (variance partitioning). There are some strong patterns in the residuals shown in Figure 3 which would benefit from further analysis and explanation. I'd also be interested to know whether the authors considered a drought severity index (e.g. scPDSI), aridity index (e.g. PANN/PET) or soil moisture index (alpha) (see e.g. Dai, 2011; Cramer and Prentice 1988). Ultimately, temperature and precipitation should interact to determine moisture stress for plant life, most notably during the drought season where the precipitation input is lowest and the evaporative demand is greatest. Conceptually at least, a moisture index might perform better than precipitation alone as a predictor for vegetation composition. In practical terms, precipitation may nevertheless be the best available predictor since the site-specific evaporative component of water stress may be poorly constrained (limited documentation of wind speeds, soil moisture capacity, etc). In some places in the text, the interpretations do not appear to take into account these real-world interactions, and the summary might be too simplistic, e.g. "the predominant driver of vegetation patterns in the Mediterranean region is water availability and not summer temperature, and this is a fact which must be borne in mind when assessing any feature in pollen-based temperature reconstructions" (p11).

I would be interested to read a little more about the training set used. Were the samples collected and analysed specifically for this study, or compiled from previous studies? Were the samples analysed by one person / one research group, or if not, how did the authors deal with harmonisation of pollen taxonomy? Are they confident that critical identifications such as evergreen vs deciduous Quercus have been made and recorded consistently? The methodology described for "averaging" across multiple moss polsters at each site is quite distinctive and I wonder whether the signal derived in this way has been evaluated in any particular previous work? This could be cited if so. I am familiar with the study of Adam and Mehringer (1975) advocating a similar averaging approach for surface soils, but not for polsters. Then, the authors cite the European Modern Pollen Database as a source of more details on the samples, but it isn't clear if data has been derived from that repository or not. Overall, these details

are important for evaluating the quality of the training set. It is good that the authors acknowledge the limitation of different sample types between the training set and the fossil material. My own previous research has been strongly critiqued in this regard, although the pragmatic necessity was similar (critique of Fletcher et al., 2010 in Birks et al., 2010). The authors here are correct that developing ideal training sets in this region remains a challenge. Finally, it would be worth commenting on why a regional training set, rather than a continental-scale training set (cf. Wei et al., 2019) was chosen to address the particular objectives of this study.

My overall impression from the seven reconstructions is one of heterogeneity in the results – and I do not find all of the synthesis statements and descriptions of the findings entirely convincing. For example, the "synchronous rise of Pann values" around 11000 cal yr BP is not so clearly evident. I think that the paper could benefit from some further approaches/efforts to illustrate the common patterns (and site specific differences) better. The use of coloured bars indicating formal stratigraphical division of the Pleistocene, Early, Mid and Late Holocene in figures 4 and 5 doesn't help the reader to visualise where each record reaches maxima, minima, etc. It might be more helpful to colour code each record into above and below average sections, $\pm 1/2$ standard deviations around the mean, for example. Going further, it might be valuable to support the written statements, for example about the declining general trend of Pann over the last 5000 years, with numerical summaries or boxplots for the different intervals to illustrate these patterns and help convince the reader that they are robustly expressed and significantly different from other intervals.

Related to this previous point, it seems that the discussion is quite rigidly organised around chronostratigraphical subdivisions, even where there is limited relevant data, e.g. one reconstruction only pertaining to the Lateglacial (cf. section 3.3.1), no signatures of the 8.2 event (cf. section 3.3.3). The authors might seek to emphasise the new contribution resulting from the new reconstructions, focusing more on both commonalities and individual features with respect to the range of values, timing of maxima and

minima, patterns of variability, etc.

Finally, I would be interested to know why the authors chose these seven sites for this study and not a wider set. While they represent a fair spatial transect across the Iberian Peninsula, the addition of further sites might certainly help discriminate the common trends and patterns (cf. comments in previous paragraphs). Due to the length of the records and chronological issues discussed, only half the records are relevant for the discussion of Lateglacial and/or earliest Holocene intervals of interest. Futhermore, only four of seven sites are located inside the Mediterranean bioclimatic zone (and only two in water-stressed settings, as per Figure 1) where precipitation is most critical for plant growth. Finally, the sites are located at different elevations from 0 to 1500 m a.s.l., which adds further complexity for the interpretation. The paper seeks to test ideas developed in works adopting a more extensive sampling approach (e.g. Mauri et al., 2015) – so it would be useful to discuss further whether the choice of seven sites here is due to a very strict set of selection criteria. Ultimately, the messages about common Holocene trends could be strengthened by additional reconstructions from other sites.

Specific comments

1. Page 1, line 19 "100% higher" – does this mean "double"?

2. Page 1, line 21-24. "In general, our results suggest that…" – the manuscript does not really return to this overarching parallel between warm high latitude and humid Iberian conditions explicitly in the discussion or conclusions. It would be good to develop this further, and consider a possible climatological mechanism for the connection.

3. Page 2, line 1 – I'm not sure the Mediterranean can itself be the transition area from Atlantic to Mediterranean – needs some rewording

4. Page 2, line 29 – give select references for the type of climate variability or events implied here

5. Page 2, line 33 – expand on "contrasting interpretations" to clarify what this study is seeking to test

6. Page 3, line 4 – can reconstructions be "fragmentary"? perhaps better "rare" or "sparse"?

7. Page 3, line 12 – the paper drifts in usage between Iberian Peninsula and Spain – better to stick to one or the other, probably IP as the geographical entity. Here, for example, the surface area of Spain as a country seems irrelevant.

8. Page 3, line19 – "south and east"?

9. Page 4, Line 19 – in Figure 1 indicate where the Eurosiberian and Mediterranean regions are.

10. Page 5, Lines 14-15 "Given that our seven pollen records. . ." – the reads as though the decision to reconstruct precipitation was a function of the availability of records, but surely the scientific aim was to reconstruct precipitation and the sites were selected accordingly? Reword according to the intended meaning.

11. Page 6 , Section 3.1 There are quite strong linear patterns in the residuals shown in Figure 3, which are not discussed in the paper – are these linked to temperature, site elevation, etc? The authors should comment on this and the implications for the reconstructions.

12. Page 7, line 13. "summer temperature records" indicate from where (geographically). . .

13. Section 3.3.1. This section is rather long relative to the amount of new contribution from the one site – can it be made more concise? The relevance of the section on Fagus on Page 8 is not clear, for example.

14. Page 8, lines 13 and 18 – what is the difference between "steppe vegetation" and "open vegetation" with respect to the implied contrast "By contrast. . ."?

15. Page 9, lines 8-13. I don't really follow how a "synchronous rise in Pann" is shown at Quintanar, San Rafael and Navarrés-3 at 11000 cal yr BP when the latter two records begin around that time and don't show any rise as such – need to clarify the key finding here

16. Section 3.3.3. It's not entirely clear to me that this section is merited. The 8.2 ka event has not been introduced in the manuscript as a particular focus of interest, and it would require careful justification in any case with respect to the sampling resolution and age uncertainties of the selected records to demonstrate that this impact could really be tested or detected with the available records. In the end there is no substantial contribution of this study in relation to this event. Regarding the point that "accurately dated high-resolution pollen records are needed" – the authors could make reference here to works by Combourieu Nebout et al. (2009) and Fletcher et al. (2010) which do identify vegetation changes in forest cover for SE Iberia around the 8.2 ka event and also give quantitative estimates for PANN anomalies associated with this events.

17. Page 10, Line 27. Given that the main finding of the Renssen et al. (2009) work cited was spatial and temporal complexity and variability in the Holocene Thermal Maximum, I think the simple equivalence of the high Pann interval in Iberia and the high latitude HTM isn't immediately comprehensible here and should be more precisely described and discussed.

18. Page 11, lines 22-28, the authors discuss human impact during the last 500 years, but then evidence agricultural activity since 7500 cal yr BP – it doesn't seem quite related in terms of the timescale or cultural setting; please clarify/prioritise whether this section is about the long record of human activity or the intensification of disturbance in recent centuries

19. Conclusions – the authors emphasise "strong spatial and latitudinal gradient during last 15 thousand years" – this gradient may be implicit but hasn't really been discussed or illustrated – this should be developed in the discussion to justify it as a conclusion.

20. Conclusions – for the finding that the "Late Pleistocene is characterized by rapid shifts in Pann values" the authors should indicate that this is based on one site only

21. Figure 1 – it would be helpful to indicate the site locations for other data presented in the paper, such as the lake records shown in Figures 6 and 7

References

Adam, D.P. and Mehringer, P.J., 1975. Modern pollen surface samples-an analysis of subsamples. Journal of Research of the US Geological Survey, 3, 733-736.

Birks, H.J.B., Heiri, O., Seppä, H. and Bjune, A.E., 2010. Strengths and weaknesses of quantitative climate reconstructions based on Late-Quaternary. The Open Ecology Journal, 3: 68-110.

Combourieu Nebout, N., Peyron, O., Dormoy, I., Desprat, S., Beaudouin, C., Kotthoff, U. and Marret, F., 2009. Rapid climatic variability in the west Mediterranean during the last 25 000 years from high resolution pollen data. Climate of the Past, 5, pp.503-521.

Cramer, W. and Prentice, I.C., 1988. Simulation of regional soil moisture deficits on a European scale, Norsk Geografisk Tidsskrift - Norwegian Journal of Geography, 42, pp. 149-151, DOI: 10.1080/00291958808552193

Dai A. 2011. Characteristics and trends in various forms of the Palmer Drought Severity Index during 1900-2008. Journal of Geophysical Research Atmospheres. 116 (D12).

Fletcher, W.J., Goñi, M.S., Peyron, O. and Dormoy, I., 2010. Abrupt climate changes of the last deglaciation detected in a Western Mediterranean forest record. Climate of the Past, 6, pp.245-264.

Mauri, A., Davis, B.A.S., Collins, P.M. and Kaplan, J.O., 2015. The climate of Europe during the Holocene: a gridded pollen-based reconstruction and its multi-proxy evaluation. Quaternary Science Reviews, 112, pp.109-127.

Wei, D., González-Sampériz, P., Gil-Romera, G., Harrison, S.P. and Prentice, I.C.,

2019. Climate changes in interior semi-arid Spain from the last interglacial to the late Holocene. Climate of the Past Discussions, https://doi.org/10.5194/cp-2019-16.

---

## Referee Comment (RC2) · Anonymous Referee #2 · 1 May 2019

Summary: The authors reconstruct mean annual precipitation for the last 15,000 years based on fossil pollen data from 7 sites in Spain. The reconstruction is based on 2 methods (WA-PLS and a Bayesian method) using a modern calibration dataset of 236 samples. They find drier conditions during the Late Glacial and late Holocene, and wetter conditions during the early-mid Holocene. This is consistent with lake level reconstructions.

The paper is well written and the methodology and approach is well established. The results are not so surprising since they have already been shown by more comprehensive pollen-based climate reconstructions by Mauri et al 2015 using MAT and Tarroso

et al 2016 using PDF method. The site of Quintanar de la Sierra was also the subject of a MAT pollen-climate reconstruction as long ago as 1997 by Penalba et al, but the authors do not appear to be aware of this. The novelty is in the use of the WA-PLS and Bayesian pollen-climate methods, which is a welcome addition to work in the region, although the calibration dataset and number of fossil pollen sites is small.

The main issues are:

1. Recognition and comparison with earlier work. This is not the first pollen-based precipitation reconstruction for the region. The authors briefly mention some of these studies, but do not make a comparison despite the fact the data is publicly available. For instance Mauri et al 2015 (https://www.ncdc.noaa.gov/paleo-search/study/18317) and Tarroso et al 2016 (doi:10.5194/cp-12-1137-2016-supplement) are both much more comprehensive in terms of the number of sites that they use in Iberia. It would be useful if the authors could compare their own work with these other studies and provide a figure which does so. On an individual site basis, it is strange that the authors also make no comment about the work of Penalba et al 1997 (doi:10.1006/qres.1997.1922) at the site of Quintanar de la Sierra. This is a much earlier pollen-based reconstruction which looks very much the same as their own reconstruction from this site.

2. Training set. The calibration data set is very small (236 samples). The study assumes that the entire Late Glacial to Holocene climate and related vegetation changes are to be found entirely in this sub-set of the current climate and vegetation of Spain. There are many thousands more modern pollen samples available from the European Modern Pollen Database, not only for Spain, but for adjoining areas which may offer more appropriate climate and vegetation analogues for the fossil samples. The authors actually cite the EMPD (Davis et al 2013), but do not provide an explanation as to why the rest of this data was not used.

3. Human impact. The authors appear to fundamentally contradict themselves when on the one hand they say how good the performance of the transfer function is when evaluated using modern pollen samples, and on the other hand they say that reconstructions based on pollen samples from the late Holocene are biased because of human impact. Presumably the modern landscape is most likely the most human impacted of the entire Holocene, so if the transfer function gives good results for the modern, why is it unable to cope with the late Holocene? The authors make a number of broad and unsubstantiated statements about biases in pollen-climate reconstructions due to human impact. These need to be toned down and contextualised with actual evidence of bias, not just evidence of human impact. The authors also need to recognise that there are many different methods and approaches to pollen-climate reconstruction, some more sensitive than others to human impact.

4. Data transparency. As a minimum, the authors need to provide enough information so that someone else could independently reproduce their study. Unfortunately, this is not currently possible with the level of information provided. The necessary information is highlighted in the detailed comments below. This information needs to be included in the supplementary data, and it includes the taxa assignments, the sites/samples and chronologies taken from the EPD and EMPD, as well as other pollen and chronological data. It would be even better if the primary data that is not already in the public domain is also included in the supplementary, or made available via a public database.

5. Evaluation. The authors compare their pollen-based precipitation reconstructions with lake level records, but the records are few, fragmentary, and much of it involves qualitative discussion. As it stands this study is relatively weak in the sense that there have already been previous pollen-based precipitation reconstructions for the region that are in many ways more comprehensive. A good way to strengthen the paper would be to provide a more comprehensive review of precipitation records in general, including lake level data. Morellon et al (2018) provide a good example of lake level synthesis which the authors partially reproduce in figure 70 p30. Morellon et al (2018) only look at the period 8-13k, but the authors here could extend this to encompass the entire Holocene. There are also plenty of lake level reconstructions that Morellon et al (2018) do not include such as Sanchez Goni, Las Pardil-las (1999) (doi: 10.1191/095968399671230625, table 4), Davis et al, Los Monegros (doi:10.1016/j.quascirev.2007.04.007, figure 8) and Reed et al, Laguna Medina (doi: 10.1191/09596830195735, figure 7).

Detailed comments:

P2, line 5-6: Mediterranean climate is specifically characterized by wet winters and dry (growing season) summers, not 'dry and wet seasons'

P2, line 21: what is a 'synthetic climate reconstruction'?

P2, line 22: there are lots of other studies you could mention..

P3, line 30: Why only 236 samples?

P3, line 31: Please provide as a minimum the condensed list of taxa that were used in the transfer function, and preferably also the full taxa list showing how all taxa were assigned to the condensed list used in the transfer function. In addition, please explain how you chose the particular taxa in the condensed taxa list used in the transfer function.

P4, line 12-13: Please provide a full list of the 236 modern pollen samples that were used, their location (lat, long and elevation) and the rainfall values assigned to these locations. A maximum of 1327mm/year seems quite low considering how wet the temperate parts of Iberia can get, and also considering the need for representative analogues for the temperate vegetation that dominated many Mediterranean areas in the early-mid Holocene.

P4, line 12-13: If samples were taken from the European Modern Pollen Database, or included in it, please include the full EMPD identity reference codes/numbers so that all of the samples can be identified. For any other samples it would be preferable if the pollen data was made available in the supplementary information or via submission to a public database or repository.

P4, line 15-19: Both Mauri et al 2015 and Tarroso et al 2016 used many more pollen sites from the region to reconstruct precipitation, since many more are available from the EPD. Can you explain why and how you chose your sites, and the basis of your reasoning for excluding the ones that you did.

P4, line 15-19: Can you identify in the table of sites, or in the supplementary information, the exact entities (you need the EPD entity reference code) and chronologies (some sites have multiple chronologies or choice of control points) you used for the EPD sites so that the primary date can be identified. For the remaining sites please specify the exact source (author or Paleodiversitas) for each of the sites in the same table. Can you also specify whether any of this data has been made public, and where this data can be downloaded. For the data that is not public, it would be preferable if it was included in the supplementary information, as well as being submitted to the EPD.

P4, line 25: 'To produce chronology' please correct the grammar

P4, line 25- P5, line 5: It would be preferable if you provided the full chronological information for each of the sites, including all control points, depths of dates, the (uncalibrated) dates themselves and their uncertainties, material dated and reference codes. Also, please say if any corrections (eg for reservoir/hardwater effects) were applied. For the EPD sites, it would be a nice gesture if you also submitted your chronologies to the EPD since they are probably better than the existing chronologies.

P5, line 7-20: You mention the complexity of the response of vegetation to climate. Is annual precipitation a reliable metric? The city of Paris gets around the same annual precipitation as Barcelona (630mm/year). Is it not growing season moisture that is more important in determining the response of vegetation to precipitation in a Mediterranean climate with hot dry summers and cool wet winters? Annual precipitation does not necessarily give any idea of the amount of growing season moisture. For instance, what would happen if the annual precipitation remained the same, but all the winter rainfall fell instead in summer? Presumably the impact on the vegetation would be

dramatic.

P5, line 19-20, P6, line 8-18: The performance metrics are only as good as the dataset that you are analyzing. If the dataset is biased (for instance, it does not sample across the entire climate space) you would not necessarily discover this from these kinds of analysis. It is not really possible to compare the performance of different datasets from different regions without standardizing the many factors that influence each dataset. In summary, r2 and rmsep are just one line of evidence for determining the performance of a transfer function, since they can easily give good results with bad data.

P6, line 20-22: Be very careful about making broad unsubstantiated statements. Evidence of human impact does not mean the same as evidence of bias. There are many different methods for reconstructing climate from pollen data, some more susceptible than others. Li et al 2014 use WA-PLS, small calibration datasets and a single site example, all of which could be expected to perform poorly in areas of heavy human impact. The main conclusion of Li et al that human impact biases the pollen-based climate reconstruction is based almost entirely on correlation, or lack of, between the pollen-based temp/precip record and other records that can anyway be expected to be different because they represent different spatial scales, temporal resolutions, or represent entirely different sensors/proxies (speleothem isotopes are a combination of precipitation and temperature signals at the destination, as well as SST and isotopic ratio of the source). Li et al also don't mention the importance of the uncertainties of the pollen-climate reconstruction in making these comparisons (one of the main effects of human impact should be to increase the uncertainties if the transfer function works correctly). Again, no one is denying that human impact can be important in pollen climate reconstructions, but you need to be careful about your evidence and phrasing here.

P6, line 26-28: Many more modern surface samples are available from the EMPD from a wide variety of depositional environments (https://epdweblog.org/european-modern-pollen-database/). I should also say, as I am sure the authors also know, that the

depositional environment also changes through time. For instance, a modern peat bog/mire may have been a lake at some time in the past, so what may make sense for the present may have to be adapted through time.

P7, line 10-11: See point 5 in the opening remarks.

P7, line 17: There are many other sites from the EPD that cover this period.

P7, line 17-28: Please read the work of Penalba et al 1997 (doi:10.1006/qres.1997.1922) at the site of Quintanar de la Sierra. The pollen-climate reconstruction in this paper looks very similar to your own.

P7, line 26-28: Be careful conflating different seasons in these comments. The Chironomid reconstruction is for summer temperatures, but your reconstruction is for annual precipitation. An increase in annual precipitation may be driven by wetter winters, unrelated to warmer summers shown by the chironomids.

P10, line 17: replace 'can be also' with 'can also be'

P10, line 23-24: This is misleading. The chironomid summer temperature record from Basa de la Mora lake by Tarrats et al 2018 indicates warmer temperatures in the early Holocene relative to the mid-late Holocene, but these temperatures were either similar or cooler than the present day. See figure 5a in Tarrats et al. The authors reconstruct a modern July air temperature of around 9.5C but they reconstruct early Holocene temperatures of around 9.1C. In fact the present July temperature for the site based on the New et al 2002 climatology adjusted for altitude is 13.2C. Tarrats et al 2018 suggest that the late Holocene samples are unreliable due to human impact, so the early Holocene summer temperatures in the chironomid reconstruction (9.1C) would in fact appear to be 4C cooler than the present day climate (13.2C).

P10, line 33: As above, Tarrats et al 2018 may SAY that their reconstruction supports Samartin et al 2017, but their reconstruction does not show this. At best, their early Holocene summer temperatures are similar to the present day if you believe the late

Holocene reconstruction (Tarrats et al, figure 5a). Or at worst, the early Holocene summer temperatures are in fact substantially cooler than modern if you follow the authors recommendations and reject the late Holocene reconstruction as unreliable.

P10, line 30-34: The Mediterranean is a big place. Samartin et al 2017 provide evidence of summer warming from 2 small adjacent lakes high in the northern Italian Apennines. This is not controversial, nor does it challenge previous work, nor should it necessarily provide support for climate models. Any analysis of multiple sites in the region will show areas of warming and areas of cooling in the early-mid Holocene. This is not surprising as it is entirely consistent with what is found in virtually every other region of the world where multi-site studies of Holocene climate have ever taken place (eg North America; Viau et al doi:10.1029/2005JD006031, Arctic; Kaufmann et al doi:10.1016/j.quascirev.2015.10.021, China; Wu et al doi: 10.1007/s00382-007-0231-3, North Atlantic; De Vernal et al doi:10.1016/j.gloplacha.2006.06.023 etc). Areas of both warming and cooling are shown in the Mediterranean in the first multi-site pollen reconstructions of summer temperatures by Huntley & Prentice (1988) (doi:10.1126/science.241.4866.687), it is shown in Wu et al (2007) (doi: 10.1007/s00382-007-0231-3) it is shown in Mauri et al (2015), based on hundreds of pollen sites from across the Mediterranean, and it is even shown in SST records from across the Mediterranean (Hessler et al 10.5194/cp-10-2237-2014). The scientific basis for a cooler early-mid Holocene is based on the regional area-average temperature calculated from the temperature trend shown at tens or hundreds of sites from across all areas of the Mediterranean (Davis et al 2003, Mauri et al 2015 etc). It is nothing new to find that individual sites and small areas show warming, it is the regional area-average that is important. This regional area-average approach is also the appropriate spatial scale to make comparisons with global climate models, in which the Mediterranean is represented by just a few grid boxes, and where countries such as Italy are barely resolved at all. For Samartin et al to truly challenge this view, or support the results of the GCMs, then they need to come up with a comparable analysis at a similar spatial scale. Even during the ubiquitous greenhouse driven global warming of the last

150 years it is still possible to find individual sites and small local areas that show cooling, but we don't then pretend that the presence of these odd sites/areas represents the regional/global picture.

P10, line 1-5: Please do not make unsubstantiated statements. Why specifically do you think that pollen-based temperature reconstructions are unreliable in the region? Please cite supporting evidence (and not other studies that make similar unsubstantiated statements such as Samartin et al 2018). Pollen-based climate reconstructions reconstruct a cooler climate in the early-mid Holocene at most (but by no means all) locations largely because of the presence of temperature vegetation (eg Prentice et al 1996 Reconstructing biomes from palaeoecological data: a general method, Collins et al 2012 doi:10.1111/j.1365-2699.2012.02738.x). The idea that these reconstructions are being driven by changes in 'Mediterranean vegetation' as the authors suggest is not therefore entirely correct. In any case the authors idea that precipitation alone drives the observed mid-early Holocene vegetation change, and not temperature, has already been tested using a process-based vegetation model in diagnostic mode by Wu et al. 2007 (doi: 10.1007/s00382-007-0231-3). Wu et al. investigated what the most probable climate would have been to generate the mid-early Holocene vegetation shown at pollen sites across the Mediterranean. They show that changes in precipitation alone cannot account for the change in vegetation, and that cooler summer temperatures are still necessary to reproduce the vegetation changes shown in the pollen record. Note that this result is also entirely independent of any potential bias that the authors suggest has been caused by later human impact on the modern vegetation. The agreement between the inverse modelling approach by Wu et al, and the calibration-based approaches by Huntley et al, Mauri et al etc, suggest that contrary to the authors suggestions, human impact is not a proven cause of bias in pollen transfer functions in the Mediterranean, at least in studies such as these that use very large training sets.

P11, line 29-33: again, please do not make broad unsubstantiated statements. Why specifically do you think that pollen-based temperature reconstructions are unreliable

in the region? Please cite supporting evidence. Evidence of human impact on vegetation is not evidence that pollen-based climate reconstructions are 'strongly influenced' by human impact. This is not to deny that the problem exists, but it is important to recognize that not all pollen-based climate reconstructions are the same, and that some have been designed specifically to limit this problem. In particular, large scale studies based on hundreds of sites and training sets of thousands of samples are arguably better able to capture the underlying regional climate signal from the noise generated by localized human impact. Other strategies include grouping taxa into common plant functional groups, and in many cases samples with high uncertainties and poor analogues that can be related to human impact are in any case filtered out. Studies based on only a few or even a single site, and those that use small training sets, are much more vulnerable. However, as the authors present themselves, transfer function performance is almost always evaluated using modern pollen samples from the modern landscape which has probably the highest human impact of the entire Holocene. The fact that we get such good performance statistics says something about the robustness of these transfer functions, although as always it is important to remember that performance statistics can be misleading. Human impact introduces noise in the relationship between vegetation and climate, usually resulting in wider uncertainties rather than simply 'wrong' values, but training set size is critical here since small datasets can be very sensitive. There are also other ways to evaluate performance. For instance, as mentioned before with the Wu et al study, cooler summer temperatures in the early-mid Holocene are also reconstructed by inverse modelling methods independent of the issue of human impact in the present-day landscape. Similarly, as the authors themselves find, the late Holocene increase in aridity reconstructed in many pollen reconstructions is also supported by lower lake levels and other independent proxies.

In any case, as here, the problem is often presented the wrong way around. The training set is tested and tuned to the modern human-impacted landscape, so it is not necessarily human impact that is the problem, but a lack of human impact in the past for which we have no modern analogue (note all calibration based methods require

modern analogues, including PDF, WA-PLS etc).

P12, line 1-7: Please refer to my previous comments. There is a world of difference between a reconstruction based on a single site or sample, and that based on many hundreds of sites or samples. The authors own transfer function has been constructed and its performance and uncertainties evaluated in the present day human impacted landscape. Human impact adds noise, but it is not necessarily overwhelming noise.

P12, line 18-19: Please be precise in your terminology. When you say 'high' do you mean higher than present during the period 8-4k? This is probably true for lake levels, but where is the evidence for higher summer temperatures? In the rest of the paper you appear to dismiss the pollen based reconstructions, so you are only left with the Pyrenees chironomid reconstruction by Tarrats et al (2018) which shows either comparable to present or most likely cooler summer temperatures. Are there other published quantitative summer temperature reconstructions from Iberia that support your conclusion?

P12, line 22-23: You conclude that late-Holocene climate reconstructions are 'thus substantially biased by human impact', but your own transfer function is based on the modern landscape which is probably the most human impacted of the entire Holocene. How can you make this statement and not negate your entire study, based on the fact that you are also applying your 'human impacted' modern calibration dataset to the earlier period? I think that you need to be more careful in making broad statements of this nature. Clearly human impact has some role, and there are certainly sites where at times the record has been influenced by human impact, sometimes significantly so. However, as I have mentioned in earlier comments, evidence of human impact on the pollen record is not evidence that climate reconstructions based on pollen data are 'thus substantially biased'. There are certainly poorly designed studies that would be susceptible to human impact, and the authors study has obvious weaknesses (small number of sites, small training set), but you should be prepared to present a much stronger argument if you want to summarily dismiss all of the previous work using

pollen-climate transfer functions in the region.

P12, line 25-29: See my opening comments. Please identify the source and identity of all of the data that you have used in the study, and preferably make available as much of the primary data that is not publicly available as possible. This can be in the form of supplementary information, or better, submit it to a recognized database (EPD, Neotoma) or data respository (Pangaea, NOAA paleoclimate). If you make data available in the supplementary information, please do so in a recognized data file format (excel, comma delimited etc), and not as a word or pdf file. I recognise that the authors have already generously submitted their data in the past, but it helps if we continue to aim for maximum data transparency in science, and especially in climate science given its high public profile.

P13, line 9-12. Please acknowledge the EPD and EMPD according to the requirements of the protocol for data use (http://europeanpollendatabase.net/datapolicy/). It is really critical that you acknowledge use of these databases so that funders can see where their money is being spent. Much of the work to support these databases is done by volunteers because of the lack of funding, and the lack of acknowledgement just makes this problem worse.

P24, Figure 1: I would recommend avoiding graded scaling, and especially multiple colours for a simple graduated scale. Contour scaling using simple 1 or 2 colour shading is a much clearer way to show this kind of information on a map. Look in any climate text book.

P27 figure 4, P28 Figure 5: I cannot understand the scaling. Along the top is every 400mm and the bottom is every 500mm. Please use the same scale, and also make the tick marks clearer. Also include a vertical line to see the anomaly from the present, not just a dot for the present precipitation. What are the 'formal sub-divisions of the Holocene'? please provide a citation. The Holocene transition seems reasonable, the end of the Laurentide ice sheet around 8k seems ok, at least in Europe directly

downstream, but what is the step-wise change in the Earths climate system at 4k? This all sounds like a hangover from the days of Blytt-Sernander and subsequent efforts to shoe-horn a Scandinavian framework onto the rest of the world.

P30, figure 7: Can you not provide a more comprehensive review of lake levels and other precipitation proxies for comparison with the pollen based reconstructions? Many are described in the text but not shown here. See #5 of my opening comments. I would also mention Harrison & Digerfeldt 1993 'European lakes as palaeohydrological and palaeoclimatic indicators' (see figure 10), which is old but still appears to be relevant today.

---

## Author Comment (AC1)

We acknowledge the detailed and constructive comments made by the reviewers. We mostly agree with the general views presented by them and we have made substantial changes to the manuscript and figures based on their suggestions. We also note that in some key questions the views presented by the reviewers differ, and we have tried to take such differing comments into account in our response in a balanced way. While revising the paper, we have also tried to control the length

5 of the paper, including the number of citations. This understandably sets some limits to our responses.

First, there are some key issues raised by all three reviewers. We will first explain how we have responded to these and then provide a point-by-point response individually to all reviewers. The reviewer comments are in blue and our responses in black.

10

**Modern pollen samples and the training set (calibration model)**

All three reviewers point out that it is important to provide a more detailed description of the modern pollen samples used for the training set (or pollen-climate calibration model) in the paper. Reviewer 2 also stresses that the modern pollen data should be made available

15 should be made available

**R 1**: "I would be interested to read a little more about the training set used. Were the samples collected and analysed specifically for this study, or compiled from previous studies? Were the samples analysed by one person / one research group, or if not, how did the authors deal with harmonisation of pollen taxonomy? Are they confident that critical identifications such as evergreen vs deciduous Quercus have been made and recorded consistently?

**R** 2: "Training set. The calibration data set is very small (236 samples). The study assumes that the entire Late Glacial to Holocene climate and related vegetation changes are to be found entirely in this sub-set of the current climate and vegetation of Spain. There are many thousands more modern pollen samples available from the European Modern Pollen Database, not only for Spain, but for adjoining areas which may offer more appropriate climate and vegetation analogues for the fossil samples. The authors actually cite the EMPD (Davis et al 2013), but do not provide an explanation as to why the rest of this data was not used..."

R 3: "However, the description of the modern pollen dataset is too short and the discussion on the multi -method
 approach needs to be improved. The discussion is essentially based on the results of the WAPLS: why? This point must be justified. If the results of the Bayesien method are not robust, then yes, you can only discuss the WAPLS, otherwise you have to discuss both.

20

25

RESPONSE: We have added a more detailed description of the modern pollen samples on page 4 in "Data sources". We explain that we selected modern pollen samples for the training set on the basis of few key criteria to make the training set as harmonized as possible. First, the procedure for sampling these selected modern pollen samples was standardized, so that all modern samples were collected by averaging the results of several moss polsters, as described in the paper. Second, all

- 5 samples in our training set were treated in the laboratory and studied under the microscope by the same person (JALS), so the harmonization of the pollen taxonomy of these 236 selected pollen samples is ensured. The identification of critical pollen morphotypes, for example those in the Quercus genus, was carried out in a consistent manner. See, for example, the papers by López-Sáez et al. (2010, 2015) in this regard.
- 10 As for the number of samples (236) in our calibrating set, we do not consider it particularly small if compared to many other regional calibration sets used in pollen-based climate reconstructions. In such a regional calibration set it is possible to better control the quality of the samples (e.g. taxonomy, sedimentary context) than in datasets including thousands of samples collected from lakes, bogs, soils samples, pollen traps etc. Moreover, the idea of using a regional calibration set is that it is designed to provide a reasonable response model for key pollen types in the calibration set and the fossil data. In other
- 15 words, the gradient of the climate variable of interest needs to be large enough to reflect the species response model. This should be the case in our Iberian calibration set, given the large gradient in precipitation from NW Spain to southern Spain. The transfer function techniques used in our paper, WAPLS and the Bayesian Bummer model, assume unimodal or Gaussian response models for the pollen types, although both are to some extent robust for other types of response models as well. Including more modern pollen samples from other regions (e.g. Central Europe or northern Europe or other continents)
- 20 would influence the optima and tolerance values of the pollen types included in the calibration model and be subsequently reflected in the climate reconstruction. It is not clear to us how this would improve the calibration set or the reconstructions in our study.

As for the comment by reviewer 2 about the availability of the modern samples ("If samples were taken from the European

- 25 Modern Pollen Database, or included in it, please include the full EMPD identity reference codes/numbers so that all of the samples can be identified. For any other samples it would be preferable if the pollen data was made available in the supplementary information or via submission to a public database or repository"), we can tell that all the samples included in this study were included in the EMPD many years ago by one of the authors (JALS). See the work of Davis et al. (2013), in which JALS is a coauthor. The majority of the 963 samples referred to Spain were contributed to the EMPD by JALS. In
- 30 short, indeed the samples used in our study come from the EMPD but at the same time correspond to our own authorship.

CHANGES: page 4 in "Data sources" we added "All pollen samples in the training set were treated in the laboratory and analyzed under the microscope by the same person (JALS), to ensure the taxonomical harmonization of these 236 selected

pollen samples. The identification of critical pollen morphotypes, for example those in the *Quercus* genus, was been carried out in a consistent manner (López-Sáez et al. (2010; 2015))."

**Selection of the climate variables:**

5

10

Second key issue raised by all reviewers is the selection of the climate variables

R 1: "The authors have chosen to focus on precipitation as the target for reconstruction and they present some good reasons why precipitation should be a strong determinant of vegetation cover across the Iberian Peninsula. However, in order to advance the aforementioned debate, it would be helpful to demonstrate that precipitation indeed does a better job than other climatic variables such as summer and winter temperature at explaining the variance in the modern pollen data (variance partitioning)...."

- R 2: "You mention the complexity of the response of vegetation to climate. Is annual precipitation a reliable metric?
  The city of Paris gets around the same annual precipitation as Barcelona (630mm/year). Is it not growing season moisture that is more important in determining the response of vegetation to precipitation in a Mediterranean climate with hot dry summers and cool wet winters? Annual precipitation does not necessarily give any idea of the amount of growing season moisture"
- 20 **R 3**: "Third point: this study does not propose a temperature reconstruction inferred from pollen: why? The authors assume that precipitation is the most important climate parameter, but this is not justified in a statistical point of view. Multivariate analyses would be required to prove the role of annual precipitation."

RESPONSE: This is indeed a crucially important question in our study, and, as a matter of fact, in all quantitative climate reconstructions. We have added some more explanation about the selection of the annual precipitation as the climate variable on page 5 ("Reconstruction of past climate variables") and added a citation to the study of Pasho et al. (2011) to the paper.

In our original paper, we acknowledged the importance of this question by stating that "The selection of the climate variable of interest is a critical step in quantitative climate reconstructions...". We also acknowledge that precipitation is not the only

30 important climate variable influencing the vegetation composition, but that temperature is also important and, in some cases, the primary determinant. We write that "However, the summer temperature may also be an important factor especially at the high altitudes (Vidal-Macua et al., 2017").

There is ample biogeographical and plant ecological evidence for the importance of precipitation for the large-scale features of vegetation composition and structure in the Iberian Peninsula. We cite some of these studies in our paper (Pasho et al. 2011; Vicente-Serrano et al. 2014; Vidal-Maqua et al. 2017), and much more have been published. Many studies have shown previously that especially in xeric Mediterranean areas tree growth is mainly limited by low precipitation, while more humid

5 regions, including the mesic Mediterranean areas and sites at higher altitude where precipitation is generally higher, the main factor constraining growth is low temperature (Vicente-Serrano, 2007; Pasho et al. 2011). Our sites are not from the high altitudes (all below 1.5 km, Table 1 in our paper), which lends support for water availability as an important variable.

As for the selection of annual precipitation instead of some other variables related to water availability, we agree with the reviewers that a more bioclimatic climate variables, such as effective precipitation or even Palmer's drought severity index, can be more informative variable. However, in the relatively small research areas such as the Iberian Peninsula, precipitation and effective precipitation are correlated and aligned with the main vegetation patterns – the highest precipitation and effective precipitation are on the W and NW coast, and the lowest in S and SE Spain. Thus the reconstructed main trends for these two correlated climate variables would be generally similar.

15

20

A statistical test about the importance of different variables in the calibration set (as suggested by R 1 and R 3) would be possible, but that would require testing many different climate variables (e.g. summer temperature, winter temperature (or temperature of the coldest month), sum of degree days, summer precipitation, winter precipitation, actual/potential evaporation ratio etc.) using for example constrained ordination techniques and hierarchical partitioning. However, exploring these questions more in detail in our paper would shift the focus of the paper and would thus require a separate paper focusing only on the modern pollen samples and the calibration set based on them.

Finally, it is useful to remind that the starting point for our paper was the fact that a number of lake-level reconstructions have been recently published from the Iberian Peninsula (see Fig 7 in our paper). Our study was designed to test and validate

25 these humidity records with pollen-based precipitation reconstructions. This does not mean that temperature would not be an important factor or that pollen-based temperature reconstructions from the region would be flawed or biased. We thus do not see any disagreement between our selection of annual precipitation as the climate variable and the views presented by the reviewers – our impression is that we all see precipitation as a critically important climate variable in pollen-based climate reconstructions in the Iberian Peninsula, but do not rule out the importance of temperature either.

CHANGES: We have added some more explanation about the selection of the annual precipitation as the climate variable on page 5 ("Reconstruction of past climate variables") and added a citation to the study of Pasho et al. (2011) to the paper.

30

The reviewers ask us to provide justification why these seven fossil records were selected for climate reconstructions

5 **R** 1: "Finally, I would be interested to know why the authors chose these seven sites for this study and not a wider set."

**R 2**: "Both Mauri et al 2015 and Tarroso et al 2016 used many more pollen sites from the region to reconstruct precipitation, since many more are available from the EPD. Can you explain why and how you chose your sites, and the basis of you reasoning for excluding the ones that you did."

**R** 3: "The choice of the 7 fossil pollen data should be explained and justified; for example, why did you only choose Quintanar de la Sierra, as the sequence which cover the Lateglacial while other pollen records are available in the EPD?"

15

20

10

RESPONSE: We have added a more detailed description on the selection of the fossil records (page 4 in "Data sources"). We now explain that the idea of this work, from its beginnings, was to make a palaeoclimatic reconstruction following a north-south transect in the Iberian Peninsula, to be able to compare humidity trends between the wetter territories of the north and what currently are semi-desert conditions in the southeast. Such a steep precipitation and humidity gradient is exclusive to the Iberian Peninsula on a European scale. Of the Iberian fossil records included in the EPD, we chose the pollen diagrams with a reasonably good chronological resolution. Moreover, the diagrams were selected so that they encompassed most of the Holocene, and if possible, Late Pleistocene with a reasonably reliable chronological control.

Of those currently available in the EPD, the seven records included are the only ones that meet these requirements. Of course, we would have liked to include key records for the knowledge of the palaeoclimate of southern Iberian Peninsula, particularly the Padul record, but unfortunately these data are not currently available. It is true that some other records could have been included, in the case of Ayoó de Vidriales or Xan de Llamas, but we believe that the chosen ones represent the Iberian climatic and biogeographical variability amply. Our study includes records near the north coast (Monte Areo), located in internal valleys also in the north (Alto de la Espina), and in northern mountains (Zalama); also in northern

30 Mediterranean mountains of the Iberian System (Quintanar de la Sierra) or in central Mediterranean mountains of the Iberian Central System (El Maíllo), in the Mediterranean region, in the peninsular east and not far from the influence of the Mediterranean sea (Navarrés), and finally in territories currently semi-desert in the southeast (San Rafael). We consider that the choice of these seven fossil records is adequate and represents adequately the climatic and biogeographical variability of the Iberian Peninsula CHANGES: page 4 in "Data sources" we added "These records were selected as they represent different climatic regions of the Iberian Peninsula from the more humid northwestern parts to the dry regions in the south."

**5 Mid-Holocene temperature trends**

In our paper, we mention the ongoing discussion about the mid-Holocene temperature patterns in the Mediterranean region. All three reviewers comment on this question:

R 1: "The paper sets out an aim of testing "contrasting interpretations" (p2, line 34) about Holocene climatic conditions, citing the works of Mauri et al., 2015 and Samartin et al., 2017. As far as I understand it, the discrepancy between these cited studies relates to the timing of maximum summer warmth during the Holocene and contrasts between previous pollen- and chironomid- reconstructions for the (western and central) Mediterranean region. I think this discrepancy should be spelled out more clearly in the introduction because some readers may not be familiar with that debate"

**R 2:** "The Mediterranean is a big place. Samartin et al 2017 provide evidence of summer warming from 2 small adjacent lakes high in the northern Italian Apennines. This is not controversial, nor does it challenge previous work, nor should it necessarily provide support for climate models..."

**R** 3: "I don't understand why the authors discuss the paper by Samartin et al, (2017) which is based on two chironomids records located at high elevation sites in northern Apennines; this paper focus on the reconstruction of the temperature of July while the aim of the paper of Ilvonen is to reconstruct precipitations. Moreover, the Tjuly reconstructed values are based on a modern chironomids dataset with only samples from Scandinavian and Alps; it's not comparable to Mediterranean taxa! So please avoid this discussion, or provide Tjuly reconstruction from pollen and discuss it."

RESPONSE: We note that here the reviewers' comment diverge, with reviewer 1 suggesting spelling out the question more clearly in the paper and reviewer 3 suggesting omitting it from the paper.

30

20

25

We write in our paper that "We did not aim to use pollen data for summer temperature reconstructions for the reasons explained earlier, and our results do not contribute directly to this debate, but in general we agree with these authors in that the predominant driver of vegetation patterns in the Mediterranean region is water availability and not summer temperature, and this is a fact which must be borne in mind when assessing any feature in pollen-based temperature

*reconstructions in this region.*" As this sentence indicates, we recognize that our data is not directly relevant for this debate, and we are certainly not trying to solve this question in our paper, mostly because we agree with R 2 who points out that the Mediterranean is a big place. Exploring this question would require a dataset that would cover all parts of the Mediterranean basin.

5

10

However, we also believe that information on temperature patterns in the Holocene are relevant when using pollen data for precipitation or lake-level data for humidity reconstructions. In response to R 3, we state that the Samartin et al. (2017) paper does not only show two chironomid-based summer temperature reconstructions from Italy, but also two model simulations for summer temperature based on two different models. Thus, the question about the mid-Holocene summer temperature conditions in the Mediterranean region is also relevant from the point of feasibility of palaeoclimate model simulations. For

these reasons, we have retained our original text about the Samartin et al. (2017) paper in our revised version.

**Earlier data**

15 All reviewers point out that there are many earlier Late Pleistocene and Holocene climate reconstructions from the Iberian Peninsula and suggest many additional papers to be cited.

**R** 1: "Regarding the point that "accurately dated high-resolution pollen records are needed" – the authors could make reference here to works by Combourieu Nebout et al. (2009) and Fletcher et al. (2010) which do identify vegetation changes in forest cover for SE Iberia around the 8.2 ka event and also give quantitative estimates for PANN anomalies associated with this events."

**R** 2: "Recognition and comparison with earlier work. This is not the first pollen-based precipitation reconstruction for the region. The authors briefly mention some of these studies, but do not make a comparison despite the fact the data is publicly available."

**R** 3: "A lot of references on past climate changes in the Mediterranean area are missing; they are needed to improve the discussion: Combourieu-Nebout et al., 2013; Bini et al., 2019; Peyron et al., 2013, Magny et al., 2013, Moreno et al., 2017...)."

30

20

25

RESPONSE: Indeed there are many earlier studies, and we have done our best to cite these papers. At the same time we have tried to keep the paper relatively short, and hence to we do aim to exhaustively cite papers from the other parts of the Mediterranean basin.

CHANGES: In response to this comment we have added references to the older papers (e.g. Combourieu-Nebout et al. 2009 on page 2). Peñalba et al. (1997) was already cited in our original paper, but we have added the following text to page 4. "The pollen data from Quintanar de la Sierra have been used earlier for quantitative climate reconstructions by Peñalba et al. (1997)".

"I'd also be interested to know whether the authors considered a drought severity index (e.g. scPDSI), aridity index (e.g. PANN/PET) or soil moisture index (alpha) (see e.g. Dai, 2011; Cramer and Prentice 1988). Ultimately, temperature and precipitation should interact to determine moisture stress for plant life, most notably during the drought season where the precipitation input is lowest and the evaporative demand is greatest. Conceptually at least, a moisture index might perform better than precipitation alone as a predictor for vegetation composition".

RESPONSE: We agree with R 1 that temperature and precipitation interact to determine the moisture stress for plant life and that these two climate variables are correlated, at least at low altitudes, so that the drought is most severe when the precipitation is low and temperature is high. However, our reconstruction extends to 15 ka, and we are hesitant to argue that this interaction and correlations extend to the Late Pleistocene period 15 ka to 12 ka. It is possible that during the Late Pleistocene the climate may have been both dry (low precipitation) and cold.

15 "The methodology described for "averaging" across multiple moss polsters at each site is quite distinctive and I wonder whether the signal derived in this way has been evaluated in any particular previous work?"

RESPONSE: As the reviewer states, averaging across multiple samples is a common procedure when surface soil samples are used for modern pollen samples. We used the method with our moss polsters to derive a pollen signal which is not too much biased by extremely local features of one moss polster. We are not aware of any evaluation of this sampling

stratigraphy with moss polsters.

"My overall impression from the seven reconstructions is one of heterogeneity in the results – and I do not find all of the synthesis statements and descriptions of the findings entirely convincing. For example, the "synchronous rise of Pann values" around 11000 cal yr BP is not so clearly evident. I think that the paper could benefit from some further approaches/efforts to illustrate the common patterns (and site specific differences) better. The use of coloured bars indicating formal stratigraphical division of the Pleistocene, Early, Mid and Late Holocene in figures 4 and 5 doesn't help the reade to visualise where each record reaches maxima, minima, etc. It might be more helpful to colour code each record into above and below average sections,  $\pm 1/2$  standard deviations around the mean, for example."

RESPONSE, CHANGES: To make the figures more informative, we have added vertical stippled lines to indicate the mean value of each reconstruction to Figs. 4 and 5.

20

25

In order to show the times with statistically significant decrease or increase in the Pann values with respect to both location and scale (meaning "level of smoothing" that is "bandwidth") we now provide a SiZer analysis by Chaudhuri and Marron (1999) for the WA-PLS reconstructions in the supplement. We have added a supplement figure (Fig. S3) which shows the SiZer maps and in supplement Table S2 we now explain the main findings from Figure S3. On page 7 we have now added

5 the following text "To show the statistically significant features in the WA-PLS reconstructions, the SiZer analysis was carried out (Fig. S3 and Table S2) (Chaudhuri and Marron, 1999)."

"Going further, it might be valuable to support the written statements, for example about the declining general trend of Pann over the last 5000 years, with numerical summaries or boxplots for the different intervals to illustrate these patterns and help convince the reader that they are robustly expressed and significantly different from other intervals."

"Related to this previous point, it seems that the discussion is quite rigidly organized around chronostratigraphical subdivisions, even where there is limited relevant data, e.g. one reconstruction only pertaining to the Lateglacial (cf. section 3.3.1), no signatures of the 8.2 event (cf. section 3.3.3). The authors might seek to emphasise the new contribution resulting from the new reconstructions, focusing more on both commonalities and individual features with respect to the range of values, timing of maxima and minima, patterns of variability, etc."

RESPONSE: Results of the SiZer analysis (Fig. S3 and Table S2) show the declining trend over the last 5000 years in four
of our records (Monte Areo, Alto de la Espina, Quintar de la Sierra and San Rafael). Furthermore, the SiZer maps also show values for maxima and minima (color changes from red to blue or from blue to red).

CHANGES: Added "Fig. S3 and Table S2" on page 11.

25 1. Page 1, line 19 "100% higher" – does this mean "double"?

**RESPONSE: Yes**

30

10

15

2. Page 1, line 21-24. "In general, our results suggest that. . ." – the manuscript does not really return to this overarching parallel between warm high latitude and humid Iberian conditions explicitly in the discussion or conclusions. It would be good to develop this further, and consider a possible climatological mechanism for the connection.

RESPONSE: Yes, but this would require using palaeoclimate modeling to understand the possible climatic mechanisms and atmospheric and oceanic processes that can explain the links between the Iberian Peninsula and the European high latitudes. In the current paper our main aim is to report the reconstructed precipitation trends, and more thorough analyses of these processes can be done in future papers in collaboration with modellers.

5

3. Page 2, line 1 - I'm not sure the Mediterranean can itself be the transition area from Atlantic to Mediterranean – needs some rewording

RESPONSE, CHANGES: Corrected to "The Iberian Peninsula is one of such regions...".

10

4. Page 2, line 29 - give select references for the type of climate variability or events implied here

RESPONSE, CHANGES: References added to Heiri et al. (2014) and Rasmussen et al. (2014)

15 5. Page 2, line 33 – expand on "contrasting interpretations" to clarify what this study is seeking to test

RESPONSE, CHANGES: We have changed "the mid-Holocene climatic conditions" to "the mid-Holocene temperature conditions". For more a thorough discussion, please see our response earlier in "Mid Holocene temperature trends".

20 6. Page 3, line 4 – can reconstructions be "fragmentary"? perhaps better "rare" or "sparse"?

RESPONSE, CHANGES: Changed to "sparse"

7. Page 3, line 12 – the paper drifts in usage between Iberian Peninsula and Spain – better to stick to one or the
 other, probably IP as the geographical entity. Here, for example, the surface area of Spain as a country seems irrelevant.

RESPONSE, CHANGES: Changed to "Iberian Peninsula" when we mean the whole region.

30 8. Page 3, line19 – "south and east"?

**RESPONSE**, CHANGES: Corrected**

9. Page 4, Line 19 – in Figure 1 indicate where the Eurosiberian and Mediterranean regions are.

RESPONSE, CHANGES: The white line in Fig. 1 indicates the boundary between these two regions (added to the figure caption).

5 10. Page 5, Lines 14-15 "Given that our seven pollen records. . ." – the reads as though the decision to reconstruct precipitation was a function of the availability of records, but surely the scientific aim was to reconstruct precipitation and the sites were selected accordingly? Reword according to the intended meaning.

RESPONSE, CHANGES: Modified to "The seven pollen records were selected from altitude lower than 1500 m a.s.l.,..."

10

11. Page 6, Section 3.1 There are quite strong linear patterns in the residuals shown in Figure 3, which are not discussed in the paper – are these linked to temperature, site elevation, etc? The authors should comment on this and the implications for the reconstructions.

- 15 RESPONSE: We explored this to some extend by checking the modern geographical location, site elevation and modern precipitation of the sites in our calibration model. We observed that most sites with biggest residual values are located at the higher end of the precipitation gradient. However, there are also some outlier sites at lower precipitation. We do not observe any specific site elevation or geographical location that would explain the outliers. Thus we cannot provide a simple explanation for the residuals but we suspect that one factor is the difficulty to obtain accurate modern precipitation values for
- 20 the sites. The other possible contributing factor is the well-documented edge effect typical to WAPLS, often leading to underestimated modern values at the higher end of the calibration models (see for example Juggins and Birks 2012).

12. Page 7, line 13. "summer temperature records" indicate from where (geographically).

25 RESPONSE, CHANGES: Changed to "...records from southwestern Europe"

13. Section 3.3.1. This section is rather long relative to the amount of new contribution from the one site - can it be made more concise? The relevance of the section on Fagus on Page 8 is not clear, for example.

30 RESPONSE, CHANGES: Shortened by deleting one sentence.

14. Page 8, lines 13 and 18 – what is the difference between "steppe vegetation" and "open vegetation" with respect to the implied contrast "By contrast. . ."?

15. Page 9, lines 8-13. I don't really follow how a "synchronous rise in Pann" is shown at Quintanar, San Rafael and Navarrés-3 at 11000 cal yr BP when the latter two records begin around that time and don't show any rise as such – need to clarify the key finding here

5

RESPONSE, CHANGES: Changed to "the high Pann values in the early Holocene around..."

16. Section 3.3.3. It's not entirely clear to me that this section is merited. The 8.2 ka event has not been introduced in the manuscript as a particular focus of interest, and it would require careful justification in any case with respect to the sampling resolution and age uncertainties of the selected records to demonstrate that this impact could really be tested or detected with the available records. In the end there is no substantial contribution of this study in relation to this event. Regarding the point that "accurately dated high-resolution pollen records are needed" – the authors could make reference here to works by Combourieu Nebout et al. (2009) and Fletcher et al. (2010) which do identify vegetation changes in forest cover for SE Iberia around the 8.2 ka event and also give quantitative estimates for PANN anomalies associated with this events.

RESPONSE: The 8.2 ka event is not a particular focus of interest in our paper, but we find it important to mention it briefly because it has been recently intensively discussed in the Mediterranean region and because some of our records indicate a slight reduction in Pann at 8400-7900 cal yr BP, possibly (but not firmly) suggesting this event in our results (e.g. Fig. 4). By stating that more high-resolution pollen records would be needed to investigate this event, we refer to studies that would particularly focus on the time period 8400-7900 cal yr BP, with sub-centennial time resolution.

Page 10, Line 27. Given that the main finding of the Renssen et al. (2009) work cited was spatial and temporal
 complexity and variability in the Holocene Thermal Maximum, I think the simple equivalence of the high Pann interval in Iberia and the high latitude HTM isn't immediately comprehensible here and should be more precisely described and discussed.

RESPONSE: We agree that such a discussion would be relevant, but it would expand the paper substantially and require incorporating the palaeoclimate model simulations in the study to understand the atmospheric and oceanic processes that can drive the temperature and precipitation trends in N and S Europe. We consider that it is best to focus on pollen-based Pann results in the current paper. 18. Page 11, lines 22-28, the authors discuss human impact during the last 500 years, but then evidence agricultural activity since 7500 cal yr BP – it doesn't seem quite related in terms of the timescale or cultural setting; please clarify/prioritise whether this section is about the long record of human activity or the intensification of disturbance in recent centuries

**5**

RESPONSE: The evidence for the beginning of agriculture in general dates to roughly 7500 cal yr BP and the text about the intense human impact on vegetation refers to the local vegetation around the Monte Areo site (We write that "*The drop of reconstructed Pann from 1500 mm to under 600 mm during the last 500 years in the Monte Areo record is an extreme example of this pattern*")

10

19. Conclusions – the authors emphasise "strong spatial and latitudinal gradient during last 15 thousand years" – this gradient may be implicit but hasn't really been discussed or illustrated – this should be developed in the discussion to justify it as a conclusion.

15 RESPONSE: The spatial and latitudinal gradient is reflected in Pann differences between the sites located in the dry regions in the South and Southeast (max. Holocene Pann in Navarrés 3 and San Rafael under 600 mm) as compared to the sites in the Eurosiberian region in the North and Northwest, with Pann over 1000 mm during the most humid periods. These features can be seen in Figs. 4 and 5.

**20 20. Conclusions – for the finding that the "Late Pleistocene is characterized by rapid shifts in Pann values" the authors should indicate that this is based on one site only**

RESPONSE: Yes, there is only one new record in our paper, but we think that in general there exist a wealth of evidence for rapid Pann or humidity changes in the Late Pleistocene in the region.

25

21. Figure 1 - it would be helpful to indicate the site locations for other data presented in the paper, such as the lake records shown in Figures 6 and 7

RESPONSE, CHANGES: We have added the site locations for the lake records shown in Fig. 7. 30

1. Recognition and comparison with earlier work. This is not the first pollen-based precipitation reconstruction for the region.

5

RESPONSE: Please see our response to general comments "Earlier data" on page 7.

However, we also agree with reviewer 1 who argues that "Also, in light of the strong environmental gradients and climatic diversity of the Iberian Peninsula, it is surprising that there are relatively few studies to date exploring climate reconstruction with this specific geographical focus."

3. Human impact.

and

4. Data transparency.

15

20

25

10

RESPONSE: Will be answered below in detailed comments.

5. Evaluation. The authors compare their pollen-based precipitation reconstructions with lake level records, but the records are few, fragmentary, and much of it involves qualitative discussion. As it stands this study is relatively weak in the sense that there have already been previous pollen-based precipitation reconstructions for the region that are in many ways more comprehensive. A good way to strengthen the paper would be to provide a more comprehensive review of precipitation records in general, including lake level data. Morellon et al (2018) provide a good example of lake level synthesis which the authors partially reproduce in figure 70 p30. Morellon et al (2018) only look at the period 8-13k, but the authors here could extend this to encompass the entire Holocene. There are also plenty of lake level reconstructions that Morellon et al (2018) do not include such as Sanchez Goni, Las Pardillas (1999) (doi: 10.1191/095968399671230625, table 4), Davis et al, Los Monegros (doi:10.1016/j.quascirev.2007.04.007, figure 8) and Reed et al, Laguna Medina (doi:10.1191/09596830195735, figure 7).

30 RESPONSE: Fig. 7 in our paper also include a lake-level curve from Lake Estanya for the whole study period (15-0 ka) from Morellon et al. (2009). The lake-level reconstructions suggested by the reviewer do not encompass the whole Holocene; The reconstruction in Davis et al. (2007) period 10,500-5500 cal yr BP and Reed et al (2001) period 9000-0 cal yr BP. Sanchez-Goni & Hannon (1999) show a table indicating periods with high and low lake level from Las Pardillas Lake, but it would be difficult to convert such data to any form of a curve with a scale. CHANGES: We note that the results of these three papers generally agree with our results, and we have added citations to them to the paper. We have modified the text (page 10 and 11) where we now write that "The more humid early-to Mid-Holocene conditions are also reported in the studies based on the two saline lakes in the Central Ebro desert (Davis &

5 Stephenson, 2007)" and "In the southern Iberian Peninsula, the lake-level reconstruction from Laguna de Medina in SW Spain suggests humidity maximum at 7000-6000, followed by a steady decline (Reed et al. 2001), while in the multi-proxy dataset from the Padul wetland in Sierra Nevada the period with highest humidity has been dated to 9500-7600 cal yr BP (Ramos-Román et al. 2018a)"

**10 **Detailed comments:**

P2, line 5-6: Mediterranean climate is specifically characterized by wet winters and dry (growing season) summers, not 'dry and wet seasons'

**15 RESPONSE, CHANGES: Corrected**

P2, line 21: what is a 'synthetic climate reconstruction'?

RESPONSE, CHANGES: Corrected (synthetic deleted)

**20**

P2, line 22: there are lots of other studies you could mention..

RESPONSE: Please see our response to general comments part "Earlier data" on page 7.

**25 P3, line 30: Why only 236 samples?**

RESPONSE: Please see our response to general comments part "Modern pollen samples and the training set (calibration model)" on pages 1-3.

30 P3, line 31: Please provide as a minimum the condensed list of taxa that were used in the transfer function, and preferably also the full taxa list showing how all taxa were assigned to the condensed list used in the transfer function. In addition, please explain how you chose the particular taxa in the condensed taxa list used in the transferfunction.

RESPONSE: All data used in our reconstructions are freely available from the authors, including a full list of taxa and the pollen and spore values. We consider this a better alternative than the taxa list only. Note that we used all terrestrial pollen and spores types (taxa) in our reconstructions.

- 5 P4, line 12-13: Please provide a full list of the 236 modern pollen samples that were used, their location (lat, long and elevation) and the rainfall values assigned to these locations. A maximum of 1327mm/year seems quite low considering how wet the temperate parts of Iberia can get, and also considering the need for representative analogues for the temperate vegetation that dominated many Mediterranean areas in the early-mid Holocene.
- 10 RESPONSE, CHANGES: The full list of modern pollen samples (lat, long, rainfall values) is already available at <a href="http://dx.doi.org/10.17632/4pznttrd4h.1">http://dx.doi.org/10.17632/4pznttrd4h.1</a>. We added the elevation information to that data. Please see also our response to general comments part "Modern pollen samples and the training set (calibration model)" on pages 1-3. Furthermore, all data used in our reconstructions are freely available from the authors.
- 15 P4, line 12-13: If samples were taken from the European Modern Pollen Database, or included in it, please include the full EMPD identity reference codes/numbers so that all of the samples can be identified. For any other samples it would be preferable if the pollen data was made available in the supplementary information or via submission toa public database or repository.
- 20 PESPONSE: Please see our response to general comments part "Modern pollen samples and the training set (calibration model)" on pages 1-3.

P4, line 15-19: Can you identify in the table of sites, or in the supplementary information, the exact entities (you need the EPD entity reference code) and chronologies (some sites have multiple chronologies or choice of control points) you used for the EPD sites so that the primary date can be identified. For the remaining sites please specify the exact source (author or Paleodiversitas) for each of the sites in the same table. Can you also specify whether any of this data has been made public, and where this data can be downloaded. For the data that is not public, it would be preferable if it was included in the supplementary information, as well as being submitted to the EPD.

30 RESPONSE: Please see our response to general comments part "Modern pollen samples and the training set (calibration model)" on pages 1-3 and "Selection of the seven fossil sites" on page 5. As we write "The chronologies of all sites are based on radiocarbon dating. In order to produce chronology for each fossil pollen record we used Bayesian age-depth model called Bchron (Haslett and Parnell, 2008). Bchron first calibrates radiocarbon dates with a calibration curve (IntCal13) and then fits the age-depth model, which is consistent with the calibrated radiocarbon dates. Assumptions for the

age-depth model are continuous, monotone and piecewise linear age-depth dependence. The age for the uppermost sediment of the core was assumed to be the year when the core was extracted. Fig. 2 shows the results of the seven Bchron runs." This means that in order to get comparable chronologies we produced chronologies for all seven pollen records based on radiocarbon date data (can be found from Table 1 references). The (median) chronologies from the Bchron runs are already

5 available at http://dx.doi.org/10.17632/4pznttrd4h.1.

**P4, line 25: 'To produce chronology' please correct the grammar**

**RESPONSE**, CHANGES: Corrected**

**10**

15

20

P4, line 25- P5, line 5: It would be preferable if you provided the full chronological information for each of the sites, including all control points, depths of dates, the (uncalibrated) dates themselves and their uncertainties, material dated and reference codes. Also, please say if any corrections (eg for reservoir/hardwater effects) were applied. For the EPD sites, it would be a nice gesture if you also submitted your chronologies to the EPD since they are probably better than the existing chronologies.

RESPONSE, CHANGES: We updated the material already provided at http://dx.doi.org/10.17632/4pznttrd4h.1. The updated material includes now all depths for each of the seven pollen record. Please see Table 1 references in order to find the radiocarbon date data for each fossil pollen record. We did not apply any corrections (eg for reservoir/hardwater effects) since it is not possible to include them to Bchron model run.

"P6, line 20-22: Be very careful about making broad unsubstantiated statements. Evidence of human impact does not mean the same as evidence of bias. There are many different methods for reconstructing climate from pollen data, some more susceptible than others. Li et al 2014 use WA-PLS, small calibration datasets and a single site example, all of which could be expected to perform poorly in areas of heavy human impact. The main conclusion of Li et al that human impact biases the pollen-based climate reconstruction is based almost entirely on correlation, or lack of, between the pollen-based temp/precip record and other records that can anyway be expected to be different because they represent different spatial scales, temporal resolutions, or represent entirely different sensors/proxies (speleothem isotopes are a combination of precipitation and temperature signals at the destination, as well as SST and isotopic ratio of the source). Li et al also don't mention the importance of the uncertainties of the pollen-climate reconstruction in making these comparisons (one of the main effects of human impact should be to increase the uncertainties if the transfer function works correctly). Again, no one is denying that human impact can be important in pollen climate reconstructions, but you need to be careful about your evidence and phrasing here.

25

P11, line 29-33: Evidence of human impact on vegetation is not evidence that pollen-based climate reconstructions are 'strongly influenced by human impact. This is not to deny that the problem exists, but it is important to recognize that not all pollen-based climate reconstructions are the same, and that some have been designed specifically to limit this problem..."

5

RESPONSE: We agree that the most broad-scale biogeographical features in the vegetation of the Iberian Peninsula, from where our training set samples come from, are controlled by climate. We can see these main features in the presence of temperate broadleaves, such as *Fagus sylvatica*, in northern Spain, while the modern pollen samples from southern Spain are

- 10 temperate broadleaves, such as *Fagus sylvatica*, in northern Spain, while the modern pollen samples from southern Spain are dominated by grasses and other more xerophytic plants. We therefore think that the main gradient in our training set reflects this climatic gradient, and this is why the performance statistics of our transfer function give reasonably high values. However, the human impact on vegetation in our study area is long-lasting, and can be locally very intensive, so that the original forest can been totally cleared for agriculture (e.g cereal or olive cultivation). The influences of such drastic human-
- 15 induced vegetation changes on pollen-based climate reconstructions are inevitable. We note, for example, that in a recent overview about the pollen data and the relative importance of climate and human impact as factors causing vegetation change in the Holocene, Roberts et al. (2019) conclude that "During the mid Holocene, most Mediterranean landscapes were transformed by a combination of climate and rural land use, but after ~3500 cal. yr BP, human actions became increasingly dominant in determining land cover".
- 20

30

We do not argue that our transfer function is particularly good when evaluated statistically. On the contrary we write that "When these performance statistics are compared with other validation tests with WA-PLS and Bayesian-based transfer functions, it can be seen that they are reasonably high, but still slightly lower than in other regional models". As for the reasonably good statistical performance of our transfer function, we state that do not know what these performance statistic would be if there were no human impact on vegetation in the areas, but very likely they would be higher.

would be if there were no human impact on vegetation in the areas, but very likely they would be hig

Later on in her/his comment, R 2 writes that "*Human impact adds noise, but it is not necessarily overwhelming noise*…" As a matter of fact, this is very much in line with our views – human impact has been intense and long-lasting in the Iberian Peninsula, and it cannot be ignored, but it does not necessarily mean that pollen records would not provide signals about the long-term climatic trends in the past.

P7, line 17: There are many other sites from the EPD that cover this period.

RESPONSE: Please see our response to general comments part "Selection of the seven fossil sites" on page 5.

P7, line 17-28: Please read the work of Penalba et al 1997 (doi:10.1006/qres.1997.1922) at the site of Quintanar de la Sierra. The pollen-climate reconstruction in this paper looks very similar to your own.

5 RESPONSE, CHANGES: Peñalba et al. (1997) was already cited in our original paper, but we have added the following text to page 4. "The pollen data from Quintanar de la Sierra have been used earlier for quantitative climate reconstructions by Peñalba et al. (1997)".

P7, line 26-28: Be careful conflating different seasons in these comments. The Chronomid reconstruction is for
 summer temperatures, but your reconstruction is for annual precipitation. An increase in annual precipitation may be driven by wetter winters, unrelated to warmer summers shown by the chironomids.

RESPONSE: We agree with this comment. See our response to the point 5 by R 2 on pages 15-16.

15 P10, line 17: replace 'can be also' with 'can also be'

**RESPONSE, CHANGES: Done**

P10, line 23-24: This is misleading. The chironomid summer temperature record from Basa de la Mora lake by
Tarrats et al 2018 indicates warmer temperatures in the early Holocene relative to the mid-late Holocene, but these temperatures were either similar or cooler than the present day. See figure 5a in Tarrats et al. The authors reconstruct a modern July air temperature of around 9.5C but they reconstruct early Holocene temperatures of around 9.1C. In fact the present July temperature for the site based on the New et al 2002 climatology adjusted for altitude is 13.2C. Tarrats et al 2018 suggest that the late Holocene samples are unreliable due to human impact, so the early Holocene summer temperatures in the chironomid reconstruction (9.1C) would in fact appear to be 4C cooler than the present day climate (13.2C).

RESPONSE, CHANGES: We agree with R 2 that in the Tarrats et al. (2018) record the reconstructed modern summer temperature is higher than during the early- to- mid-Holocene temperature maximum. We have modified the text on page 11

30 where we now write that "In addition, palaeoclimate reconstructions from central Pyrenees, based on chironomids, and thus independent of pollen data, indicate that the summer temperature was high from 8800 to 6200 cal yr BP, although stilll lower than the modern summer temperature at the site (Tarrats et al., 2018)"

We have deleted the Tarrats et al. (2018) curve from Fig. 7.

P12, line 18-19: Please be precise in your terminology. When you say 'high' do you mean higher than present during the period 8-4k? This is probably true for lake levels, but where is the evidence for higher summer temperatures? In the rest of the paper you appear to dismiss the pollen based reconstructions, so you are only left with the Pyrenees chironomid reconstruction by Tarrats et al (2018) which shows either comparable to present or most likely cooler summer temperatures. Are there other published quantitative summer temperature reconstructions from Iberia that support your conclusion?

RESPONSE: See our response to the point 5 by R 2 on pages 15-16.

10

5

P13, line 9-12. Please acknowledge the EPD and EMPD according to the requirements of the protocol for data use (http://europeanpollendatabase.net/datapolicy/).

RESPONSE, CHANGES: Acknowledgements were added to the paper.

15

25

P24, Figure 1: I would recommend avoiding graded scaling, and especially multiple colours for a simple graduated scale. Contour scaling using simple 1 or 2 colour shading is a much clearer way to show this kind of information on a map. Look in any climate text book.

**20 RESPONSE, CHANGES: Done**

P27 figure 4, P28 Figure 5: I cannot understand the scaling. Along the top is every 400mm and the bottom is every 500mm. Please use the same scale, and also make the tick marks clearer. Also include a vertical line to see the anomaly from the present, not just a dot for the present precipitation What are the 'formal sub-divisions of the Holocene'? please provide a citation.

RESPONSE, CHANGES: We have modified the scaling. The Pann scale is now indicated clearly for each panel in the figure. We have enlarged the tick marks.

30 The vertical line has been added. It shows the anomaly from the mean (not the present precipitation). The dot indicates the modern measured precipitation value in each reconstruction location.

The formal subdivisions of the Holocene, ratified by the International Commission on Stratigraphy (ICS) in 2018, are Greenlandian, Northgrippian and Megalayan. These subdivision and their age limits can be checked for the homepage of the ICS. We have added this explanation to the figure caption and added a citation to Walker et al. (2018).

- 5 "P30, figure 7: Can you not provide a more comprehensive review of lake levels and other precipitation proxies for comparison with the pollen based reconstructions? Many are described in the text but not shown here. See #5 of my opening comments. I would also mention Harrison & Digerfeldt 1993 'European lakes as palaeohydrological and palaeoclimatic indicators' (see figure 10), which is old but still appears to be relevant today."
- 10 RESPONSE: Please see the response to opening comment 5 above on pages 15-16.

"I first suggest to better highlight the innovative side of this study. In particular if we compare the objective of this work with those of the paper of Tarroso et al (2016) which focus on the reconstruction of the climate (Temperature and precipitation) in Iberian peninsula during the last 15000 years from pollen data. This study has several positive points that should be further highlighted in the text: a new modern pollen dataset, a multi-method approach... "

RESPONSE: We agree that our precipitation reconstructions partly differ from those in Tarroso et al. (2016). It is more difficult for us to assess the reasons of these differences. In essence, it would be necessary to re-run the reconstructions of
Tarroso et al. (2016) to find out what are the factors mostly contributing to the differences. We consider such an assessment outside the focus on our paper.

"However, the description of the modern pollen dataset is too short and the discussion on the multi –method approach needs to be improved. The discussion is essentially based on the results of the WAPLS: why? This point must be justified. If the results of the Bayesien method are not robust, then yes, you can only discuss the WAPLS, otherwise you have to discuss both."

and

15

5

"The WAPLS is a classic method, often used in paleoclimate studies. In contrast, the Bayesien reconstruction method is newer: could you better explain and justify the choice of this method instead or other more classical methods (PDF, MAT, PSL...). More references on the Bayesien reconstruction method are required: where this method has been tested and applied? For which time periods?"

RESPONSE, CHANGES: We agree with the reviewer that it is good to use more than one reconstruction technique in a paper such as this. In response to this comment, we now write on page 6 that "*The Bayesian modelling provides some potential advantages, such as joint inference and a clearer modelling of uncertainty, in climate reconstructions (Parnell et al. 2016)*".

As R 3 mentions WA-PLS method is often used in paleoclimate studies. However, in recent years, the popularity of 30 Bayesian modelling in paleoclimate reconstructions has increased and there are also user-friendly Bayesian models available like R package Bclim (Bayesian Palaeoclimate Reconstruction from Pollen Data, Parnell et al., 2015) and BUMBER (Bayesian user-friendly model for palaeo-environmental reconstruction, Holden et al., 2017). For more discussion about Bayesian models and comparison between Bayesian methods and other methods see for example Birks et al., 2010; Holden et al., 2017; Li et al., 2016 and Parnell et al. 2016. In order to see the compatibility of the results based on the two reconstruction techniques we performed correlation analysis using scale space multiresolution correlation analysis (Pasanen and Holmström, 2016). As we now write in the Supplement "Scale space multiresolution correlation analysis takes into consideration the possibility that the correlation between two

- 5 time series may change over time and can have different features when inspected at different time scales. The method has two steps. In the first step the time series are decomposed into a number of scale-dependent components and in the second step the local temporal changes in correlation between pairs of such components are explored by using weighted correlation within a sliding time window of varying length." As a result method identifies the time intervals and the time scales for which correlation is credibly positive or negative. We have added a supplement figure (Fig. S4) which shows the results
- 10 from the scale space multiresolution correlation analysis. These results suggest that we obtain similar reconstructed features with multiple timescales for Monte Areo, Quintanar de la Sierra, San Rafael and El Maíllo with both methods. The features for Zalama and Navarrés-3 are similar in long timescales and for Alto de la Espina the reconstructed features seem to differ.

As regards the compatibility of the results based on the two reconstruction techniques, we have added the following text on page 7: "According to the results of the scale space multiresolution correlation analysis (Pasanen and Holmström, 2016), Alto de la Espina is the only one where the main features of the two reconstruction techniques are different (Fig. S4)."

In the figures we have chosen to show only the WA-PLS reconstructions in order to keep the figures clear and simple. The WA-PLS reconstructions were chosen based on the slightly higher performance statistics for the modern pollen-climate training set.

20

25

"My second point concerns the lack of comparison of your results with the precipitation curves available in the Mediterranean area: the study of Tarroso et al (2016) for Spain, Dormoy et al (2009) for south Spain; studies of Peyron et al. (2011; 2013), Combourieu-Nebout et al., 2013 and Magny et al (2013) for Italy. It's important to add these curves in the figures (6 or 7?) to discuss the regional climate pattern. Particularly the curves of Tarroso et al., (2016) which are based on another climate reconstruction method, the PDF, show clearly a different pattern than the precipitation reconstructed here; the differences have to be discussed more in depth."

RESPONSE, CHANGES: This is a justified argument. It would indeed be interesting to compare more extensively our 30 results with various types of precipitation and water availability related proxy records from the Iberian Peninsula and the western and central Mediterranean regions. The reason we have not done so, is, understandably, that such comparisons would make the paper much longer and figures much larger. For this reason, we have, for example, shown lake-level curves only from the Iberian Peninsula, and not from Italy or other regions, in our Fig. 7. We agree that our precipitation reconstructions partly differ from those in Tarroso et al. (2016). It is more difficult for us to assess the reasons of these differences. In essence, it would be necessary to re-run the reconstructions of Tarroso et al. (2016) to find out what are the factors mostly contributing to the differences. We consider such an assessment outside the focus on our paper.

5

10

In response to this comment we have added references to the older papers (e.g. Combourieu-Nebout et al. 2009 on page 2).

"Much of the discussion and figures are based on chironomidss temperature curves (figures 6, and7), so either the authors remove the temperature curves to base the discussion only on precipitation (and compate it with more regional precipitation patterns), or the authors apply their methods to produce temperature curves, or the authors include the temperature curves of Tarroso et al for Spain."

RESPONSE, CHANGES: Chironomid-based temperature reconstruction is now deleted from Fig. 7. See our response to R 2 on page 20.

**15**

"Last point: Authors don't investigate the links of these reconstructed climate changes with the different climate forcings. It's an important missing point."

RESPONSE: To explore and understand the influence of different forcings on Pann trends in the Iberian Peninsula would require the use of palaeoclimate models. The aim of our paper is to report and discuss the pollen-based reconstructions results. The study involving model simulations and discussing the role of different forcing factors can be done as a next step in the future.

**Other points**

**25**

**Data sources**

30

"The paragraph on the modern pollen dataset is too short given that the quality and accuracy of the modern pollen dataset is very important in transfer functions. The modern pollen dataset used here has never been published, so more details are needed: could you add a table or a map with the biome corresponding to each modern sample? We need it to be sure that all the vegetation type occurring in the past are included in your dataset. I particularly think about the more herbaceous during the Younger Dryas and the taxa of the Bolling/Allerod. Another important point to discuss is the human impact: how do you deal with that in the modern dataset? Do you exclude anthropic taxa?"

RESPONSE, CHANGES: We have now added more information about the modern pollen dataset, see our response to general comments on page 1-3. We also discuss the role of human impact, see our response to R 2 on page 18-19. We did not exclude any terrestrial pollen or spore taxa from our dataset, because defining the anthropogenic taxa would be more or less subjective and thus would add a source of subjectivity to our transfer function.

5

Reconstruction of past variables

"Line 26: you test the performance of the calibration of the transfer function, you don't test the performance of the modern training set: please correct."

10

RESPONSE, CHANGES: Corrected by deleting "to test the performance of our modern pollen-climate training set and"

"Line 28: reformulate: for constructing the transfer functions for annual precipitation"

15 RESPONSE, CHANGES: We changed "for constructing" to "to calculate".

Results and Discussion, Transfer function performance

"The fig 3 (observed/reconstructed) is not discussed at all in the text. More sentences are needed to comment the performance of each method, for example: some high precipitation values are clearly underestimated with the WAPLS: why? May be these samples are biased by human impact and could be considered as outliers and then removed from the dataset."

RESPONSE: We explored the residuals in Fig 3 to some extend by checking the modern geographical location, site 25 elevation and modern precipitation of the sites in our calibration model. We observed that most sites with biggest residual values are located at the higher end of the precipitation gradient. However, there are also some outlier sites at lower precipitation. We do not observe any specific site elevation or geographical location that would explain the outliers. Thus we cannot provide a simple explanation for the residuals but we suspect that one factor is the difficulty to obtain accurate modern precipitation values for the sites. The other possible contributing factor is the well-documented edge effect typical to

30 WAPLS, often leading to underestimated modern values at the higher end of the calibration models (see for example Juggins and Birks 2012). There is no clear evidence that sites with biggest residuals should be considered as outliers (for example due to human impact) and therefore removed.

"Line 20-23: The R2 for PANN is always lower than the R2 for temperature; I don't understand why the authors compare their R2 with R2 in China; more European or Mediterranean calibrations are available (check the bibliography: Bordon et al., 2009...). It will also be important to test the spatial autocorrelation see the papers by Telford and Birks, 2009 and others), to evaluate the performance of the models, did you do it? "

**5**

RESPONSE: Yes, we actually did this, but we did not include the h-block tests in the original paper because of lack of space. The result of this test indicates some spatial autocorrelation in the calibration set. For example, when we set h value as 20 km RMSEP increases to 170.7595 mm and R2 decreases to 0.4803. With this 20 km radius an average of 5.5 sites (min=1, max=20) were omitted in the h-block runs.

10

CHANGES: In the revised version of the paper, we have added the following text "We also tested our WA-PLS calibration model for the possible spatial autocorrelation using the h-block test (Telford and Birks 2009). When the h value is set at 20 km, RMSEP increases to 170 mm and R2 decreases to 0.48. With this 20 km radius an average of 5.5 sites (min=1, max=20) were omitted in the h-block runs. This indicates some spatial autocorrelation in our calibration model. This is probably

15 inevitable in a dataset such as ours, which is based on moss polster samples often collected from sites near each other."

Results and Discussion, Evaluation of the reconstructions

"Line 5: please reformulate: some differences in the levels of reconstructed Pann values"

**20**

RESPONSE, CHANGES: Changed to "the actual levels of reconstructed Pann values may differ to some extend".

Precipitation trends

**25 "Replace Late Pleistocene by Lateglacial"**

RESPONSE: We have retained the term "Late Pleistocene" because it is an ICS ratified term and in line with the terminology we use for the subdivision of Holocene in our paper.

30 "Line 19, 21: Pann is not a record, it's a reconstructed value, clarify"

**RESPONSE, CHANGES: Done**

"It's hard to see on the figures 4, 5 the climate patterns discussed in the text. For example: line 21 " show an increasing trend between 14500 and 14250 cal BP". The scale of the figure is not adapted to follow the discussion. Please correct."

5 RESPONSE, CHANGES: To make the figures more informative, we have added vertical stippled lines to indicate the mean value of each reconstruction to Figs. 4 and 5. We have modified the scaling. The Pann scale is now indicated clearly for each panel in the figure. We have enlarged the tick marks.

"P. 8, line 3: I don't agree with the author's interpretation: the pattern at Q de la Sierra is not stable and is not in agreement with a relatively stable rainfall pattern in northern Iberian Peninsula during the younger dryas."

RESPONSE: We use the expression "*a relatively stable rainfall pattern*" which we think is justified. There is less variability in the GS-1 stadial sequence than in the early Holocene in the Quintanar de la Sierra record.

15 "Line 24: The comparison with the lake levels is hard to follow; Estanya lake DOESN'T reflect large climate changes during Younger Dryas (fig 7): correct it."

RESPONSE, CHANGES: Corrected. We now write that "In the Estanya Lake record, at lower altitude in the Pyrenees, the reconstructed lake level is low during the GS-1, but drops even lower at the GS-1-Holocene transition (Fig. 7)"

20

10

"P.9, line 2 "... lower during the period 12.900 to 11.700 cal BP": to be nuanced; it depends where in Spain: in Villarquemado, Fuentillo and Padul, the lake levels were high during the Younger Dryas."

RESPONSE: OK, we mean the lake levels in N Spain, cited in our paper

25

"P9, lines24-26, need to better explain the reasons of differences between proxies reconstructions: precipitation seasonality..., check the papers by Magny et al 2013 and others"

and

30

"8.2 ka event: discussion on this major event is too short, check the references on the 8.2 ka event (Magny et al 2003, 2013...) to improve the discussion"

RESPONSE: As already answered to R 1 the 8.2 ka event is not a particular focus of interest in our paper, but we find it important to mention it briefly because it has been recently intensively discussed in the Mediterranean region and because some of our records indicate a slight reduction in Pann at 8400-7900 cal yr BP, possibly (but not firmly) suggesting this event in our results (e.g. Fig. 4). By stating that more high-resolution pollen records would be needed to investigate this event, we refer to studies that would particularly focus on the time period 8400-7900 cal yr BP, with sub-centennial time

resolution.

5

**"P11, line 4: I don't see where are the "reasons explained earlier"**

- 10 RESPONSE: On page 2 we say that our study was designed to test and validate humidity records with pollen-based precipitation reconstructions "Given the steep gradients and the coupling between vegetation and water availability, past vegetation changes in the Iberian Peninsula provide a means to investigate past water availability and precipitation changes. In the last decades, a number of studies based on lake level, pollen, and speleothem data have dealt with synthetic climate reconstructions in the Iberian Peninsula (Tarroso et al., 2016; Morellón et al., 2018). Here, we report pollen-based
- 15 quantitative precipitation reconstruction results based on a transfer function approach from seven pollen records from different parts of the Iberian Peninsula following a North-South transect from the Atlantic to the Mediterranean climatic domain. To provide a regional synthesis of the precipitation and humidity changes, we compare our pollen-based precipitation reconstructions with independent records of humidity, such as lake-level data from the Iberian Peninsula, from ~15,000 calibrated years before present (cal yr BP) to the present."
- 20

"P11, line 14: do you take into account Pteridium in our dataset? I don't think so, so may be exclude these samples."

RESPONSE: *Pteridium* is included in our calibration model and in the reconstructions because the purpose of our study was to include all terrestrial pollen and spore types in the reconstructions.

Figures

"figure1: the two regions Eurosiberian and Mediterranean must be indicated on the map"

30

RESPONSE, CHANGE: The white line in Fig. 1 indicates the boundary between these two regions. Added to the figure caption.

**"fig2: may be better in supplementary material"**

RESPONSE: We decided to keep the Fig. 2 in its original place in order to emphasize that we produced comparable chronologies for all seven cores based on radiocarbon date data.

5 "fig 3: not discussed in the text, to be done; check the outliers and remove it if they are linked to human impact"

RESPONSE: Already answered above. Please see the answer on page 26.

"fig 4 and 5: to discuss the climate trends, you have to you trace the figures in anomalies (differences between pastand 0k value) to avoid altitude bias."

RESPONSE: Already answered above. Please see the answer on page 28.

"For clarity, I strongly recommend to the authors to merge the figures 4 and 5, and to put on the same graphs the curves obtained with both methods (all in anomalies)."

RESPONSE: We considered this but found it more informative to show these two reconstruction outputs separately.

"The different chronozons must appeared on the figures: GS1(or Y Dryas), Holocene to help to follow the discussion."

RESPONSE, CHANGES: Added to Figs. 6 and 7.

"fig 7: The different chronozons must appeared on the figures: GS1(or Y Dryas),in dot..."

**25**

RESPONSE, CHANGES: Added

**References used only in the responses to reviewers**

Holden, P., Birks, H.J.B., Brooks, S., Bush, M., Hwang, G., Matthews-Bird, F., Valencia, B., and van Woesik, R.: BUMPER v1. 0: a Bayesian user-friendly model for palaeo-environmental reconstruction, Geosci.Model.Dev, 10(1), 483-498, 2017.

30 Parnell, A., Sweeney, J., Doan, T.K., Salter-Townshend, M., Allen, J.R., Huntley, B., and Haslett, J.: Bayesian inference for palaeoclimate with time uncertainty and stochastic volatility, J.Roy.Stat.Soc.C-App, J ROY STAT SOC C-APP, 64(1), 115-138, 2015. Vicente-Serrano, S.M.: Evaluating the impact of drought using remote sensing in a Mediterranean, semi-arid region. Nat.Hazards, 40, 173–208, 2007.

**References added to the article**

Chaudhuri, P., and Marron, J.S.: SiZer for exploration of structures in curves. J.Am.Stat.Assoc, 94(447), 807-823, 1999.

5 Combourieu-Nebout, N., Peyron, O., Dormoy, I., Desprat, S., Beaudouin, C., Kotthoff, U., and Marret, F.: Rapid climatic variability in the west Mediterranean during the last 25 000 years from high resolution pollen data, Clim.Past, 5, 503-521, 2009.

Fang, K., Chen, D., Ilvonen, L., Frank, D., Pasanen, L., Holmström, L., Zhao, Y., Zhang, P., and Seppä, H.: Time-varying relationships among oceanic and atmospheric modes: A turning point at around 1940, Quatern.Int., 487, 12-25, 2018.

10 Parnell, A., Haslett, J., Sweeney, J., Doan, T., Allen, J., and Huntley, B.: Joint palaeoclimate reconstruction from pollen data via forward models and climate histories, Quaternary.Sci.Rev., 151, 111-126, 2016. Pasanen, L., and Holmström, L.: Scale space multiresolution correlation analysis for time series data, Computational Statistics, 32(1), 197-218, 2017.

Pasho, E., Camaero, J.J., de Luiz, M., and Vicente-Serrano, S.M.: Impacts of drought at different time scales on forest growth across a wide climatic gradient in north-eastern Spain, Agr.Forest.Meteorol., 151, 1800-1811, 2011.

Roberts, N., Woodbridge, J., Palmisano, A., Bevan, A., Fyfe, R., and Shannon, S: Mediterranean landscape change during the Holocene: Synthesis, comparison and regional trends in population, land cover and climate, The Holocene, 29, 923-937, 2019.

Telford, R.J., and Birks, H.J.B.: Evaluation of transfer functions in spatially structured environments, Quaternary.Sci.Rev., 28, 1309–1316, 2009.

Walker, M., Head, M.J., Berkelhammer, M., Björck, S., Cheng, H., Cwynar, L., Fisher, D., Gkinis , V., Long, A., Lowe, J., Newnham, R., Olander Rasmussen, S., and Weiss, H.: Formal ratification of the subdivision of the Holocene Series/ Epoch (Quaternary System/Period): two new Global Boundary Stratotype Sections and Points (GSSPs) and three new stages/ subseries, Episodes, 41, 213-223, 2018.

25

15

**Quantitative reconstruction of precipitation changes in the Iberian Peninsula during the Late Pleistocene and the Holocene**

Liisa Ilvonen1,2, José Antonio López-Sáez3, Lasse Holmström4, Francisca Alba-Sánchez5, Sebastián Pérez-Díaz3, José S. Carrión6, Heikki Seppä1

[revised manuscript text omitted]

- 15 2017; Vidal-Macua et al., 2017). However, the summer temperature may also be an important factor especially at more mesic sites and at the high altitudes (Pasho et al., 2011; Vidal-Macua et al., 2017). It is realistic to accept that no single climatic variable can account for the complete influence of climate on vegetation and that no single or few reconstructed climate variables can capture the full spectrum climate patterns and changes in the past. Given that our The seven pollen records were selected are from sites located at from altitude lower than 1500 m a.s.l., and the climate variable we have reconstructed is
- 20 annual mean precipitation (Pann). In our study region, Pann is an ecologically important and conceptually simple variable, which can be used in comparisons with other palaeoclimate records and model simulations. Precipitation has a clear zonal pattern in the Iberian Peninsula, and its importance for vegetation patterns is reflected by the comparable zonality of vegetation. In the leave-one-out cross-validation test, Pann has high r2 and low RMSEP (Table 2), demonstrating that it accounts for a large proportion of variance in the precipitation-related climatic patterns in the region.
- 25 We use two different, complementary quantitative techniques, weighted-averaging partial least squares regression technique (WA-PLS) and Bayesian modelling, to test the performance of our modern pollen climate training set and to produce the past precipitation reconstructions from the seven pollen records. With both techniques, all 236 modern pollen samples were used for constructing to calculate the transfer functions for modern annual precipitation (Pann). WA-PLS is a non-linear, unimodal regression and calibration technique commonly used in quantitative environmental reconstructions (Juggins and Birks, 2012).
- 30 In all cases, we used a two-component WA-PLS model by ter Braak and Juggins (1993). Training set pollen data values (as percentages) were square root transformed for WA-PLS regression in order to reduce noise in the data. Calculation of WA-PLS transfer functions was performed in the C2 programme (Juggins, 2007).
The Bayesian modelling provides some potential advantages, such as joint inference and a clearer modelling of uncertainty, in climate reconstructions (Parnell et al. 2016). The Bayesian reconstruction method used is based on Bummer, a Bayesian hierarchical multinomial regression model introduced in Vasko et al. (2002). In the basic Bummer model, the observed pollen taxon relative abundances are modelled by a multinomial distribution, where the taxon occurrence probabilities are treated as

5 Dirichlet-distributed random variables whose distribution is determined by the pollen environmental response parameters as well as the mean annual precipitation. The taxon environmental response is modelled by a unimodal Gaussian function, with shape and mean determined by the response parameters alpha (scale), beta (optimal precipitation) and gamma (tolerance); See Figure S2. The prior distributions of the model parameters are listed in Table S1.

[revised manuscript text omitted]
., 2009, 2018). The more humid early-to Mid-Holocene conditions are also reported in the studies based on the two saline lakes in the Central Ebro desert (Davis & Stephenson, 2007). In the southern Iberian Peninsula, the lake-level reconstruction from Laguna de Medina in SW Spain suggests humidity maximum at 7000-6000, followed by a steady decline (Reed et al. 2001), while in the multi-proxy dataset from the Padul wetland in Sierra Nevada the period with highest humidity has been dated to 9500-7600 cal yr BP (Ramos-Román et al. 2018a).

5 In addition, palaeoclimate reconstructions from central Pyrenees, based on chironomids, and thus independent of pollen data, indicate that the summer temperaturtemperature was es were high from 8800 to 6200 cal yr BP, although still lower than the modern summer temperature at the site (
[revised manuscript text omitted]

5

10